# Overriding FUS autoregulation in mice triggers gain-of-toxic dysfunctions in RNA metabolism and autophagy-lysosome axis

Shuo-Chien Ling[1,2,3,4,5,6]*, Somasish Ghosh Dastidar[7], Seiya Tokunaga[1], Wan Yun Ho[4], Kenneth Lim[4], Hristelina Ilieva[1,2]†, Philippe A Parone[1,2]‡, Sheue-Houy Tyan[3,8], Tsemay M Tse[4], Jer-Cherng Chang[1], Oleksandr Platoshyn[9], Ngoc B Bui[1], Anh Bui[1], Anne Vetto[1], Shuying Sun[1,2]§, Melissa McAlonis-Downes[1,2], Joo Seok Han[1,2]#, Debbie Swing[10], Katannya Kapeli[4], Gene W Yeo[2,4,7], Lino Tessarollo[10], Martin Marsala[9], Christopher E Shaw[11,12], Greg Tucker-Kellogg[13], Albert R La Spada[2,3,7]¶, Clotilde Lagier-Tourenne[1,3]**, Sandrine Da Cruz[1], Don W Cleveland[1,2,3]*

*For correspondence:
shuochien@gmail.com (S-CL);
dcleveland@ucsd.edu (DWC)

Present address: †Saint Luke's Neurological Institute, Neuromuscular Division, Kansas City, United States; ‡Fate Therapeutics, San Diego, Uited States; §Department of Pathology, Johns Hopkins University, Baltimore, United States; #Neuracle Genetics, Seoul, Korea; ¶Departments of Neurology, Neurobiology, and Cell Biology, Duke Center for Neurodegeneration & Neurotherapeutics, Duke University School of Medicine, Durham, United States; **Department of Neurology, Massachusetts General Hospital, Charlestown, United States

[1]Ludwig Institute for Cancer Research, University of California, San Diego, San Diego, United States; [2]Department of Cellular and Molecular Medicine, University of California, San Diego, San Diego, United States; [3]Department of Neurosciences, University of California, San Diego, San Diego, United States; [4]Department of Physiology, National University of Singapore, Singapore, Singapore; [5]Neurobiology/ Ageing Programme, National University of Singapore, Singapore, Singapore; [6]Program in Neuroscience and Behavior Disorders, Duke-NUS Medical School, Singapore, Singapore; [7]Sanford Consortium for Regenerative Medicine, University of California, San Diego, San Diego, United States; [8]Department of Medicine, National University of Singapore, Singapore, Singapore; [9]Department of Anesthesiology, University of California, San Diego, San Diego, United States; [10]Mouse Cancer Genetics Program, National Cancer Institute, Frederick, United States; [11]Dementia Research Institute Centre, Maurice Wohl Clinical Neuroscience Institute, Institute of Psychiatry, Psychology and Neuroscience, King's College London, London, United Kingdom; [12]Centre for Brain Research, University of Auckland, Auckland, New Zealand; [13]Department of Biological Sciences, National University of Singapore, Singapore, Singapore

**Abstract** Mutations in coding and non-coding regions of FUS cause amyotrophic lateral sclerosis (ALS). The latter mutations may exert toxicity by increasing FUS accumulation. We show here that broad expression within the nervous system of wild-type or either of two ALS-linked mutants of human FUS in mice produces progressive motor phenotypes accompanied by characteristic ALS-like pathology. FUS levels are autoregulated by a mechanism in which human FUS downregulates endogenous FUS at mRNA and protein levels. Increasing wild-type human FUS expression achieved by saturating this autoregulatory mechanism produces a rapidly progressive phenotype and dose-dependent lethality. Transcriptome analysis reveals mis-regulation of genes that are largely not observed upon FUS reduction. Likely mechanisms for FUS neurotoxicity include autophagy inhibition and defective RNA metabolism. Thus, our results reveal that overriding FUS autoregulation will trigger gain-of-function toxicity via altered autophagy-lysosome pathway and RNA metabolism function, highlighting a role for protein and RNA dyshomeostasis in FUS-mediated toxicity.
DOI: https://doi.org/10.7554/eLife.40811.001

## Introduction

Two seemingly different adult-onset neurodegenerative diseases, amyotrophic lateral sclerosis (ALS) and frontotemporal degeneration (FTD), share overlapping clinical and pathological characteristics (reviewed in *Ling et al., 2013*; *Robberecht and Philips, 2013*; *Gao et al., 2017*). The landmark discovery of TAR-DNA binding protein (TDP-43) as a major component of ubiquitinated aggregates in ALS and FTLD (frontotemporal lobar degeneration, a term used to stress the pathological classification of FTD) patients (*Neumann et al., 2006*; *Arai et al., 2006*) led to the identification of ALS and FTD-causing mutations in the genes encoding two nucleic acid-binding proteins, TDP-43 (*Sreedharan et al., 2008*; *Kabashi et al., 2008*; *Van Deerlin et al., 2008*) and fused in sarcoma/translocated in liposarcoma (FUS/TLS) (hereafter referred to as FUS) (*Kwiatkowski et al., 2009*; *Vance et al., 2009*). In the past decade, mutations in several additional genes, including GGGGCC hexanucleotide-repeat expansion in the *C9ORF72* gene (*DeJesus-Hernandez et al., 2011*; *Renton et al., 2011*; *Gijselinck et al., 2012*) and point mutations in *UBQLN2* (*Deng et al., 2011*), *VCP* (*Johnson et al., 2010*), *CHMP2B* (*Momeni et al., 2006*; *Parkinson et al., 2006*), and *TBK1* (*Cirulli et al., 2015*; *Freischmidt et al., 2015*; *Pottier et al., 2015*) were also identified as genetic causes for both ALS and FTD. These genetic discoveries, coupled with pathological inclusions of TDP-43 (*Neumann et al., 2006*; *Arai et al., 2006*) or FUS (*Neumann et al., 2009*) that are found both in ALS and FTD, have supported common molecular mechanisms, in particular, disruption in RNA and protein homeostasis, to underlie both diseases (reviewed in *Ling et al., 2013*; *Lattante et al., 2015*; *Taylor et al., 2016*).

Molecularly, FUS is a 526 amino acid protein containing a prion-like low-complexity domain (*Kato et al., 2012*; *Cushman et al., 2010*), followed by a nuclear export signal, a RNA recognition motif (RRM) domain, arginine/glycine (R/G)-rich domains, a zinc-finger motif and nuclear localization signal. FUS binds to single- and double-stranded DNA as well as RNA and participates in multiple cellular functions (*Ling et al., 2013*; *Tan and Manley, 2009*; *Lagier-Tourenne et al., 2010*; *Schwartz et al., 2015*; *Ling, 2018*), in particular in transcription-splicing coupling (*Lagier-Tourenne et al., 2012*; *Yu and Reed, 2015*), alternative splicing and polyadenylation (*Lagier-Tourenne et al., 2012*; *Ishigaki et al., 2012*; *Rogelj et al., 2012*; *Sun et al., 2015*; *Masuda et al., 2015*; *Reber et al., 2016*), and the localization and translation of RNA (*Kanai et al., 2004*; *Fujii and Takumi, 2005*; *Yasuda et al., 2013*). A preponderance of the ALS/FTD causing mutations (48 out of 60) is clustered in the FUS extreme C-terminus that contains its non-canonical nuclear localization signal (known as PY-NLS) (*Dormann et al., 2010*; *Lattante et al., 2013*). Correspondingly, such FUS mutants have been shown to result in increased cytosolic accumulation which correlates with disease severity (*Dormann et al., 2010*; *Bosco et al., 2010*; *Gal et al., 2011*; *Vance et al., 2013*). Neuronal cytoplasmic inclusions of FUS are found in ALS patients with mutations in FUS (*Kwiatkowski et al., 2009*; *Vance et al., 2009*; *Dormann et al., 2010*), suggesting that (*i*) disturbing the nuclear-cytosolic distribution can lead to FUS proteinopathy, and (*ii*) loss of nuclear RNA processing functions may contribute to ALS pathogenesis. Intriguingly, mutations in the 3'-UTR of FUS leading to elevated FUS accumulation are known to be causal for ALS (*Sabatelli et al., 2013*). This is reminiscent of increased copy number of the *APP* gene (which encodes amyloid precursor protein and is causal for Alzheimer's disease [*Sleegers et al., 2006*]) and of the *SNCA* gene (which encodes α-synuclein and is causal for Parkinson's disease [*Singleton et al., 2003*]). Therefore, the evidence suggests elevated expression levels of genes encoding these pathological hallmarks are sufficient to drive neurodegeneration.

The pathological hallmark of FUS inclusions in ALS and FTD is characterized by the loss of nuclear FUS immunoreactivity with concomitant cytosolic accumulation, suggesting that both loss of nuclear FUS function and gain of additional toxic properties may be involved. However, Kino and colleagues used outbred FUS knockout mice to show that these FUS knockout mice do not develop ALS disease phenotypes (*Kino et al., 2015*). In contrast, mice expressing disease-causing FUS mutations or FUS with NLS-deletions developed motor neuron degeneration, favoring a 'gain-of-toxic properties' model (*Scekic-Zahirovic et al., 2016*; *Sharma et al., 2016*; *Shiihashi et al., 2016*; *Devoy et al., 2017*; *López-Erauskin et al., 2018*). Indeed, ALS-linked mutations in FUS have been suggested to affect diverse functions, including gain and loss of RNA processing (*Sun et al., 2015*; *Reber et al.,*

2016), deregulation of SMN function (*Sun et al., 2015*; *Yamazaki et al., 2012*; *Groen et al., 2013*; *Tsuiji et al., 2013*), biogenesis of circular RNA (*Errichelli et al., 2017*), DNA damage response (*Qiu et al., 2014*; *Wang et al., 2013*), axonal transport (*Groen et al., 2013*; *Guo et al., 2017*), activity-dependent translation (*Sephton et al., 2014*), and intra axonal protein synthesis (*López-Erauskin et al., 2018*). Although all of these defects may contribute to ALS and FTD pathogenesis, how any of these proposed dysfunctions contributes to neurodegeneration remains to be determined.

Autophagy, a tightly regulated catabolic process, degrades long-lived proteins, membrane proteins, and organelles via the lysosome (*Mizushima and Komatsu, 2011*; *Shen and Mizushima, 2014*). Not surprisingly, autophagy dysfunction has been implicated in various neurodegenerative diseases, including ALS (*Wong and Cuervo, 2010*; *Nixon and Yang, 2012*). The notion that autophagy dysfunction contributes to ALS-FTD pathogenesis is strongly supported by the identification of numerous ALS-FTD genes involved in autophagy regulation, including *SQSTM1* (*Fecto et al., 2011*), *OPTN* (*Maruyama et al., 2010*; *Wong and Holzbaur, 2014*), *TBK1* (*Cirulli et al., 2015*; *Freischmidt et al., 2015*; *Pottier et al., 2015*), and *VCP* (*Johnson et al., 2010*). Intriguingly, recent work has shown that loss of TDP-43 inhibits autophagy by blocking the fusion of autophagosomes with lysosomes (*Xia et al., 2016*), and that enhancing autophagy may be beneficial for mice modeling TDP-43 proteinopathies (*Wang et al., 2012*). Stress granules are cleared by autophagy (*Buchan et al., 2013*), further supporting the link between RNA-containing ribonucleoprotein complexes and autophagy.

Here we report that widespread expression of wild type FUS or ALS-linked mutations (R514G and R521C) in FUS within the central nervous system of mice can cause progressive motor deficits accompanied by ALS-like lower motor neuron pathology. An increase in expression of wild type FUS sharply accelerates disease phenotype and triggers early mortality accompanied by disturbances in both protein homeostasis and RNA processing, likely by saturating FUS autoregulation. Furthermore, we report that increased expression of wild type or disease-linked mutant FUS inhibits autophagy, suggestive of a potential gain-of-function proteotoxicity stress mechanism contributing to FUS-mediated neurodegeneration.

## Results

### Generation of 'floxed' FUS transgenic mice with broad expression in the central nervous system

Transgenic mice were produced to express either wild-type or either of two ALS-linked mutant human FUS broadly throughout the central nervous system (CNS) using the murine prion promoter (*Arnold et al., 2013*) previously reported to drive transgene expression most abundantly in the CNS, including neurons, astrocytes and oligodendrocytes (see below). cDNAs for human wild type (hereafter referred as $FUS^{WT}$) and either of two ALS-linked mutants of FUS, R514G (arginine to glycine substitution at amino acid 514, hereafter referred as $FUS^{R514G}$) and R521C (arginine to cysteine substitution at amino acid 521, hereafter referred as $FUS^{R521C}$) were fused to a N-terminal hemagglutinin (HA) tag and placed under the control of the murine prion promoter (*Figure 1A*). Each transgene was flanked with loxP sites to permit deletion in the presence of Cre recombinase activity.

Twelve lines were established from 34 founders (three wild type lines from seven founders, 5 R514G lines from 13 founders, and 4 R521C lines from 14 founders). The human FUS transgene (detected with an antibody to the HA-tag) was mostly confined to the CNS [with little or no expression in other tissues (*Figure 1—figure supplement 1*)], in a pattern mimicking endogenous FUS in nuclei of NeuN-positive neurons (*Figure 1B–C*), GFAP-positive astrocytes (*Figure 1—figure supplement 2A*), and CC1-positive oligodendrocytes of spinal cords (*Figure 1—figure supplement 2B*) as well as in most regions of the brain (*Figure 1—figure supplement 3*). The three FUS transgenes ($FUS^{WT}$, $FUS^{R514G}$ and $FUS^{R521C}$) accumulated to 0.5 to 1.5-fold the level of mouse FUS (*Figure 1—figure supplement 1A–B*) (as determined by immunoblotting brain extracts using an antibody that recognizes human and mouse FUS protein with comparable affinity [*López-Erauskin et al., 2018*]). Endogenous mouse FUS was reduced both at the protein (*Figure 1—figure supplement 1A–B*) and mRNA levels to 30–60% of non-transgenic level (*Figure 1—figure supplement 1B and D*) in all six established transgenic lines, consistent with an autoregulatory mechanism acting at the FUS RNA

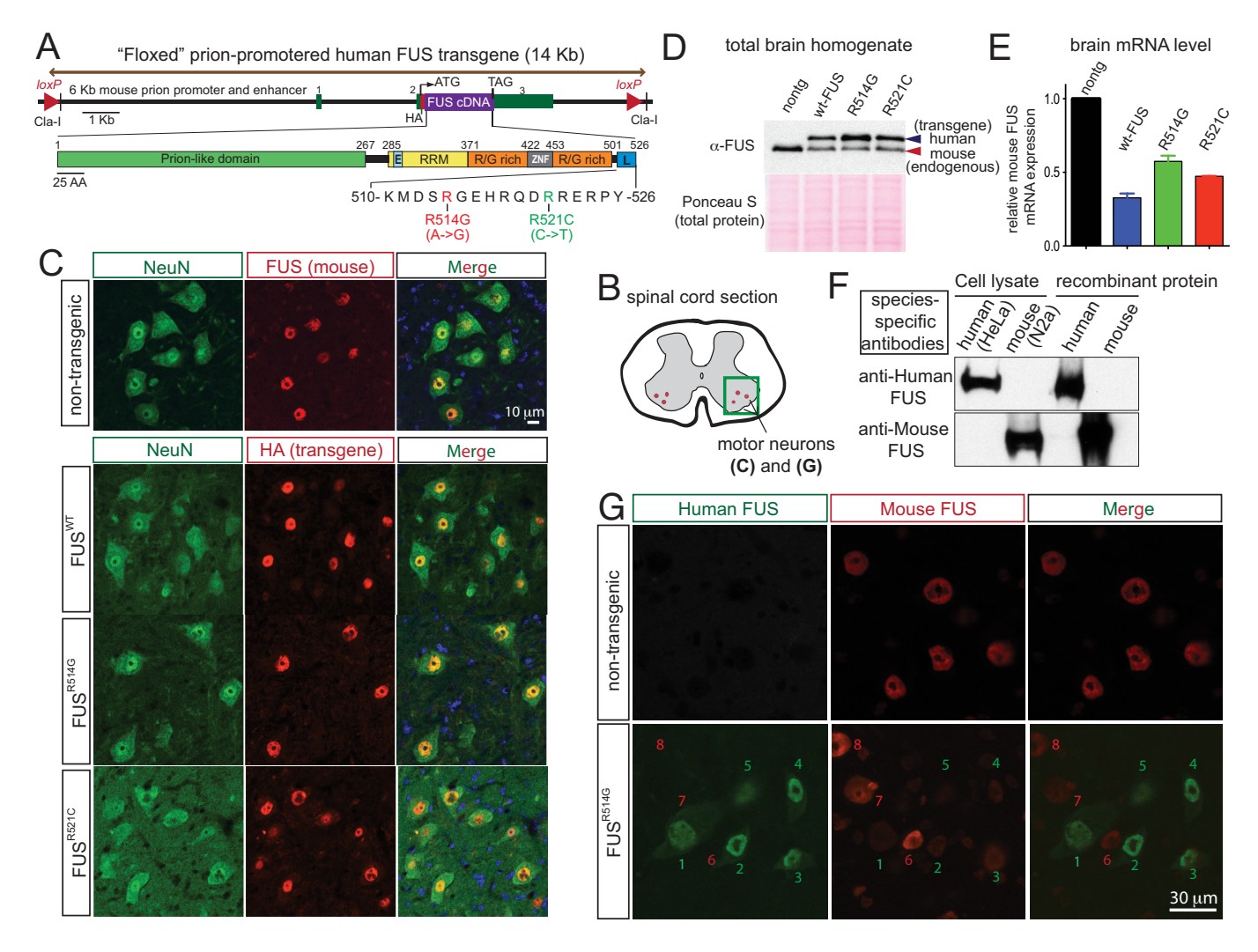

**Figure 1.** Autoregulation of FUS level in adult mouse nervous system. (**A**) Schematic representation of transgene constructs for wild type and mutant FUS using the murine prion promoter. Human cDNAs encoding wild type or R514G or R521C mutants of FUS were N-terminally fused to an HA tag and inserted between non-coding exons 2 and 3 of mouse prion gene. The mutations used in this study are within the nuclear localization signal as indicated. The constructs were flanked by the loxP sites of the same orientation. (**B**) Schematic of spinal cord sections indicates the ventral horn regions that were shown in (**C**) and (**G**). (**C**) Double immunostaining of spinal cord ventral horns of 2-month-old animals with an anti-FUS (red) or anti-HA antibody (red) and antibodies for markers of neurons (NeuN, green). Both endogenous mouse FUS and human FUS proteins are localized to the neuronal nucleus. (**D**) Immunoblots of total brain homogenates from the selected mouse lines expressing wild type and mutant FUS. Blue arrows denote the human FUS and the red arrows denote the endogenous mouse FUS. Total proteins stained with Ponceau S were used to show equal loading. (**E**) Relative mRNA levels of the endogenous FUS mRNA measured by RT-qPCR. Consistent with protein level, mRNA showed ~50% reduction of endogenous FUS. The data represent the average value from three animals per line ±SEM. (**F**) Immunoblots of species-specific antibodies for either human- or mouse-specific FUS. Cell lysates from human (HeLa) and mouse (Neuro2A) cell lines and purified recombinant human and mouse FUS were probed with either human or mouse-specific FUS antibodies. (**G**) Double immunostaining of spinal cord ventral horns of 2-month-old *prnp*-FUS[R514G] animals with an anti-HA antibody (green, transgene) and anti-mouse FUS (red, endogenous) antibody showed that the human transgenes were able to down-regulate the endogenous mouse FUS level.

DOI: https://doi.org/10.7554/eLife.40811.002

The following figure supplements are available for figure 1:

**Figure supplement 1.** Tissue expression pattern of endogenous FUS and human transgene in mice.

DOI: https://doi.org/10.7554/eLife.40811.003

**Figure supplement 2.** Expression pattern of FUS in mouse spinal cord.

DOI: https://doi.org/10.7554/eLife.40811.004

*Figure 1 continued on next page*

*Figure 1 continued*

**Figure supplement 3.** Human FUS transgenes are expressed in most regions of the brain including the cerebellum, cortex, hippocampus and striatum of transgenic mice at 2 months of age.

DOI: https://doi.org/10.7554/eLife.40811.005

levels. Lines with most comparable levels of wild type (line 101 for FUS$^{WT}$) and mutant (line 124 for FUS$^{R514G}$ and line 135 for FUS$^{R521C}$) human and mouse FUS proteins (*Figure 1D,E*) were selected for further characterization.

To further confirm the autoregulatory pathway at a single cell level, we exploited antibodies we have generated that are specific to either human or mouse FUS (*Figure 1F*) to distinguish either protein using confocal microscopy. Spinal cord motor neurons expressing higher levels of human FUS (cells labeled as 1, 2, 3, 4, five in *Figure 1G*) had near complete elimination of mouse FUS, while the subset of neurons without detectable human FUS retained a level of mouse FUS indistinguishable from that in mouse neurons of non-transgenic animals (cells labeled as 6, 7, eight in *Figure 1G*). Collectively, we conclude that an auto-regulatory mechanism regulating FUS mRNA stability or maturation to maintain FUS level within individual cells of the mouse central nervous system (CNS).

## Age-dependent lower motor deficits in FUS transgenic mice

Although transgenic mice from all three genotypes appeared normal at birth and developed normal weight into adulthood, all FUS transgenic animals expressing wild type or ALS-linked mutations in which human FUS accumulated to at least half the FUS level in normal mice developed age-dependent abnormal posture, with lower stance and hunched back, an abnormal clasping response, reduced hind limb spread (upper panel of *Figure 2A*), and a progressively abnormal gait (lower panel of *Figure 2A*). While all genotypes displayed normal stride-length at two-months of age, shorter front and hind-limb stride-length developed by 12 months of age (*Figure 2B*) in all FUS transgenic lines.

To further examine whether the age-dependent motor deficits in the FUS transgenic mice were accompanied by neuromuscular abnormalities similar to those clinically observed in human ALS, electromyograms (EMG) were recorded from the gastrocnemius muscle in the absence of any neurogenic stimulus (in isoflurane-anesthetized animals) (*Figure 2C*). Consistent with our previous findings (*Arnold et al., 2013*), high-frequency spontaneous firings of the motor units (i.e., fibrillations) were recorded in symptomatic SOD1$^{G93A}$ transgenic mice that will ultimately develop fatal paralytic ALS-like disease. Similar fibrillations, albeit less frequent, were observed in all 12-month-old FUS transgenic mice, which were absent in non-transgenic littermates (*Figure 2C*), indicating widespread denervation of neuromuscular junctions (NMJs) and motor unit degeneration and regeneration.

The degree of denervation of the gastrocnemius muscle was further evaluated by colocalization of markers to both presynaptic terminals and postsynaptic densities with synaptophysin and α-Bungarotoxin, respectively. While muscles were fully innervated at 2 months of age in all FUS transgenic mice, up to ~30% of the NMJs were lost by 12 months of age (*Figure 2D*), consistent with the abnormalities detected by EMG. This was accompanied by an age-dependent loss of large (>8.5 μm) caliber motor axons in 12-month-old FUS$^{WT}$ and FUS$^{R514G}$ mice compared to non-transgenic littermates (*Figure 2E*; *Figure 2—figure supplement 1*), although no significant loss of motor axons was observed in aged FUS$^{R521C}$ mice. Sensory axons of the L5 root were lost in an age-dependent manner in all three transgenic FUS lines (18 ~ 26% reduction by 12 months of age) compared to non-transgenic controls (*Figure 2F*; *Figure 2—figure supplement 2*). Finally, immunofluorescence analysis of the mouse lumbar spinal cords revealed age-dependent loss of motor neurons (*Figure 2G*; *Figure 2—figure supplement 3A*) and increased astrogliosis (*Figure 2—figure supplement 3B*) in 12-month-old transgenic FUS$^{R514G}$, FUS$^{R521C}$ and FUS$^{WT}$ mice, compared to non-transgenic littermates. Taken together, expression of human FUS at levels between 0.7 and 1.2 of the normal mouse FUS level produces age-dependent, selective denervation and degeneration of lower motor axons.

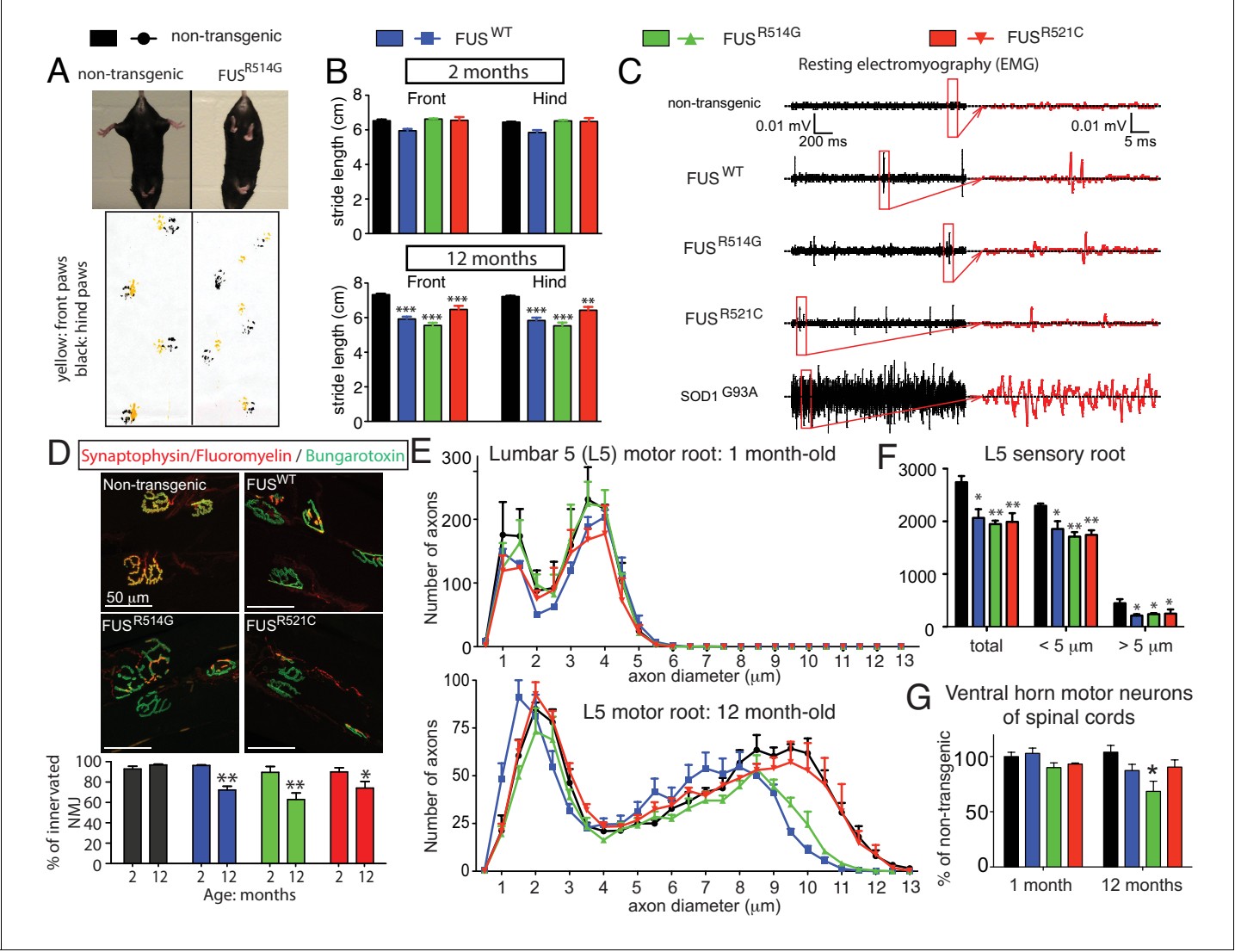

**Figure 2.** Age-dependent, mutant-enhanced toxicity in *prnp*-FUS transgenic mice. (**A**) Representative images of an 8-month-old FUS[R514G] transgenic mice and a littermate control revealing reduced hindlimb spread and clasping in the mutant mouse. Representative trace of footprint analysis for testing gait abnormality is shown in the lower panel. The front and hind paws were coated with yellow and black paint, respectively. (**B**) Statistical analysis of front and hind-limb stride length in FUS transgenic mice. The data represent an average of at least four animals ± SEM (***: p<0.001, **: p<0.01). (**C**) Resting EMG recording from the gastrocnemius muscle in isoflurane-anesthetized animals in the absence of any stimulus. (**D**) Double staining of neuromuscular junctions in the gastrocnemius of 12-month-old animals using anti-synaptophysin (red), Fluoromyelin red (red) and α-Bungarotoxin Alexa 488 (green) (top panel). Quantification of the percentage of innervated neuromuscular junctions in the gastrocnemius of FUS transgenic animals and their control littermates (see counting criteria in Materials and methods). The data represent the average of at least three animals ± SEM. (**: p<0.01, *: p<0.05). (**E**) Distributions of axonal diameters in motor axons of the L5 lumbar motor root in 1- (top panel) or 12- (lower panel) month-old animals. Data points represent the averaged distribution of axon diameters from the entire root of at least three mice for each genotype and age group. (**F**) Quantification of the average number of sensory axons: total, below or above 5 μm caliber diameter, in 12-month-old animals. The data represent the average of at least three animals ± SEM. (**G**) Quantification of the number of ChAT positive neurons in the ventral horn of spinal cords from FUS transgenic animals and their control littermates (n ≥ 3).

DOI: https://doi.org/10.7554/eLife.40811.006

The following source data and figure supplements are available for figure 2:

**Source data 1.** Characterization of lower motor neuron system in prnp-FUS mice.

DOI: https://doi.org/10.7554/eLife.40811.007

**Figure supplement 1.** Progressive degeneration of L5 motor roots in *prnp*-FUS mice.

DOI: https://doi.org/10.7554/eLife.40811.008

**Figure supplement 2.** Progressive loss of L5 sensory roots in *prnp*-FUS mice.

*Figure 2 continued on next page*

*Figure 2 continued*

DOI: https://doi.org/10.7554/eLife.40811.009

**Figure supplement 3.** Expression of hFUS causes astrogliosis and a reduction in the number of ChAT neurons in the ventral horn of lumbar spinal cord.
DOI: https://doi.org/10.7554/eLife.40811.010

## Acute sensitivity to the level of FUS: elevated level of human wild-type FUS accelerates motor neuron disease signs and causes early lethality

To test if higher levels of FUS could be achieved by overriding the underlying autoregulatory mechanism through increased expression of the FUS transgene, we next generated homozygotes (line 101) or double hemizygotes (line 101 crossed with line 136) of FUS^WT mice which were obtained at a normal Mendelian ratio. No homozygous animals were obtained for FUS^R521C (despite production of multiple litters), suggesting that ALS-linked mutation in FUS may be more toxic than the wild type FUS. To rule out potential deleterious effects caused by insertion of the FUS transgene in the coding region of a critical gene, the integration sites were identified for two FUS^WT and two FUS^R521C lines (see Supplemental information for experimental details). In line 136 of FUS^WT, the transgene integrated between protein-coding exon 12 and 13 of the *Mkl1* gene, while in line 101 expressing FUS^WT the transgene integrated between the noncoding exon 1 and 2 of *Inpp4b*, encoding type II inositol-3,4-bisphosphate 4-phosphatase. For line 136 FUS^R521C, the transgene landed in an intergenic region on chromosome 17, while for line 135 FUS^R521C the transgene inserted at the 3'-UTR of an uncharacterized protein (*Naaladl2* gene).

As predicted, a dose-dependent down-regulation of endogenous FUS and increased accumulation of total FUS exceeding twice the protein level that found in non-transgenic spinal cord homogenates was achieved in the viable doubly hemizygous (line 101 crossed with line 136), and homozygous (line 101) FUS^WT animals compared to singly hemizygous (line 101 and 136) FUS^WT transgenic littermates (*Figure 3A*). The dose-dependent down-regulation of mouse FUS was further confirmed at the mRNA level (*Figure 3B*).

While the doubly hemizygous (line 101 crossed with line 136), and homozygous (line 101) FUS^WT transgenic mice appeared normal at birth, they rapidly developed neurological disease as early as 2 weeks of age, including abnormal clasping (upper panel of *Figure 3C*) and abnormal gait (lower panel of *Figure 3C*). A marked reduced stride length of the front and hind limbs of both doubly hemizygous and homozygous FUS^WT mice was detected by 3 weeks of age (*Figure 3D*), which was accompanied by progressive worsened clasping, lowered posture and increased tremor when compared with the age-matched non-transgenic littermates and singly hemizygous FUS^WT mice (*Figure 3E*). These doubly hemizygous and homozygous FUS^WT mice eventually developed paralysis. None survived beyond 40 days of age (*Figure 3F*). Despite transgene integration between the noncoding exon 1 and 2 of *Inpp4b* gene in line 101 of FUS^WT mice, INPP4B immunoblotting revealed unaltered levels of INPP4B in comparing non-transgenic, hemizygous and homozygous FUS^WT mice (*Figure 3—figure supplement 1*), ruling out the possibility that any aspect of the phenotype in homozygotes of this line was caused by reduction in INPP4B.

Doubly hemizygous and homozygous FUS^WT mice developed by 30 day of age drastic (~92%) denervation of neuromuscular junctions (NMJs) of the gastrocnemius muscle when compared with the age-matched non-transgenic littermates and singly hemizygous FUS^WT mice (*Figure 4A–B*). This severe denervation was accompanied by significant degeneration and loss of motor axons (*Figure 4C–D*) and spinal cord motor neurons in 30-day-old doubly hemizygotes (29% and 35%, respectively) and homozygotes of FUS^WT mice (19% and 23%, respectively) compared to non-transgenic littermates or singly hemizygous FUS^WT mice (*Figure 4E*). Sensory axons of the L5 root were also lost (~30% reduction) in doubly hemizygous and homozygous FUS^WT mice (*Figure 4—figure supplement 1*). Despite near absence of both endogenous mouse and human FUS accumulation in microglia (<5% of cells are co-labeled with Iba-1 staining) (*Figure 5D*), a significant increase in microgliosis and astrogliosis (scored with Iba-I and GFAP immunoreactivity, respectively) in the lumbar spinal cords of doubly hemizygous and homozygous FUS^WT mice (*Figure 5*; *Figure 5—figure supplement 1*) was observed. Altogether, increased accumulation of FUS to 2-fold above its normal level is sufficient to produce age-dependent, fatal motor neuron disease.

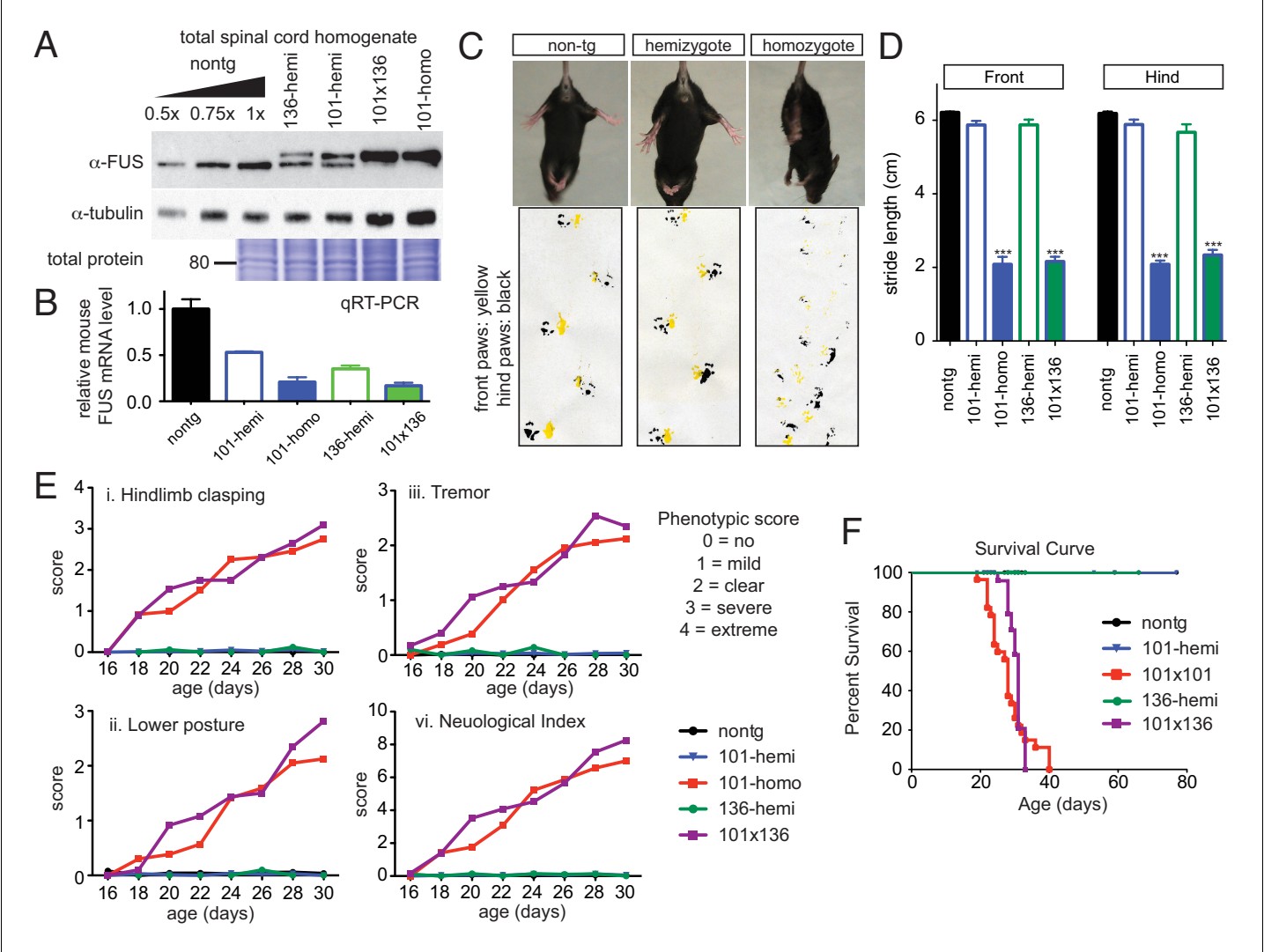

**Figure 3.** Acute sensitivity to FUS level. (**A**) Immunoblots of total whole spinal cord homogenate from non-transgenic, singly transgenic (line 101 or line 136), doubly transgenic (line 101 and line 136), and homozygous (line 101) of FUS$^{WT}$ mice. Dilutions of homogenates from non-transgenic animals were used to assess the reduction of endogenous FUS. Both tubulin immunoblot and Coomassie blue staining of total proteins were examined to ensure equal loading. (**B**) Relative mRNA levels of the endogenous FUS mRNA measured by RT-qPCR. Consistent with protein level, mRNA showed inverse reduction of endogenous FUS with increasing human FUS transgene. The data represent the average value from three animals per line ±SEM. (**C**) Hindlimb clasping phenotypes (upper panel) and footprint-gait analysis (lower panel) of littermates from non-transgenic, transgene hemizygote (line 101), and transgene homozygote (line 101) of FUS$^{WT}$ mice. (**D**) Quantifications of footprint analysis showed significant reduction in the stride length in doubly transgenic (line 101 and line 136) and transgene homozygote (line 101), but not in non-transgenic and singly transgenic animals. (**E**) Progressive neurological phenotype in FUS over-expressing mice. Hindlimb clasping, lower posture and tremor phenotypes were scored based on the severity of the phenotype from postnatal day 16 onward (n > 5 per data point). Additions of all three measurements were plotted as neurological index. Only FUS overexpressing (doubly transgenic and transgene homozygote) mice develop progressive phenotypes. (**F**) Survival curve of non-transgenic, singly transgenic (line 101 or line 136), doubly transgenic (line 101 and line 136), and homozygous (line 101) of FUS$^{WT}$ mice. None of the FUS-overexpressing mice survived more than 40 days.

DOI: https://doi.org/10.7554/eLife.40811.011

The following source data and figure supplement are available for figure 3:

**Source data 1.** Gait analysis for FUS transgenic mice.

DOI: https://doi.org/10.7554/eLife.40811.012

**Figure supplement 1.** Dose-dependent down-regulation of FUS expression.

DOI: https://doi.org/10.7554/eLife.40811.013

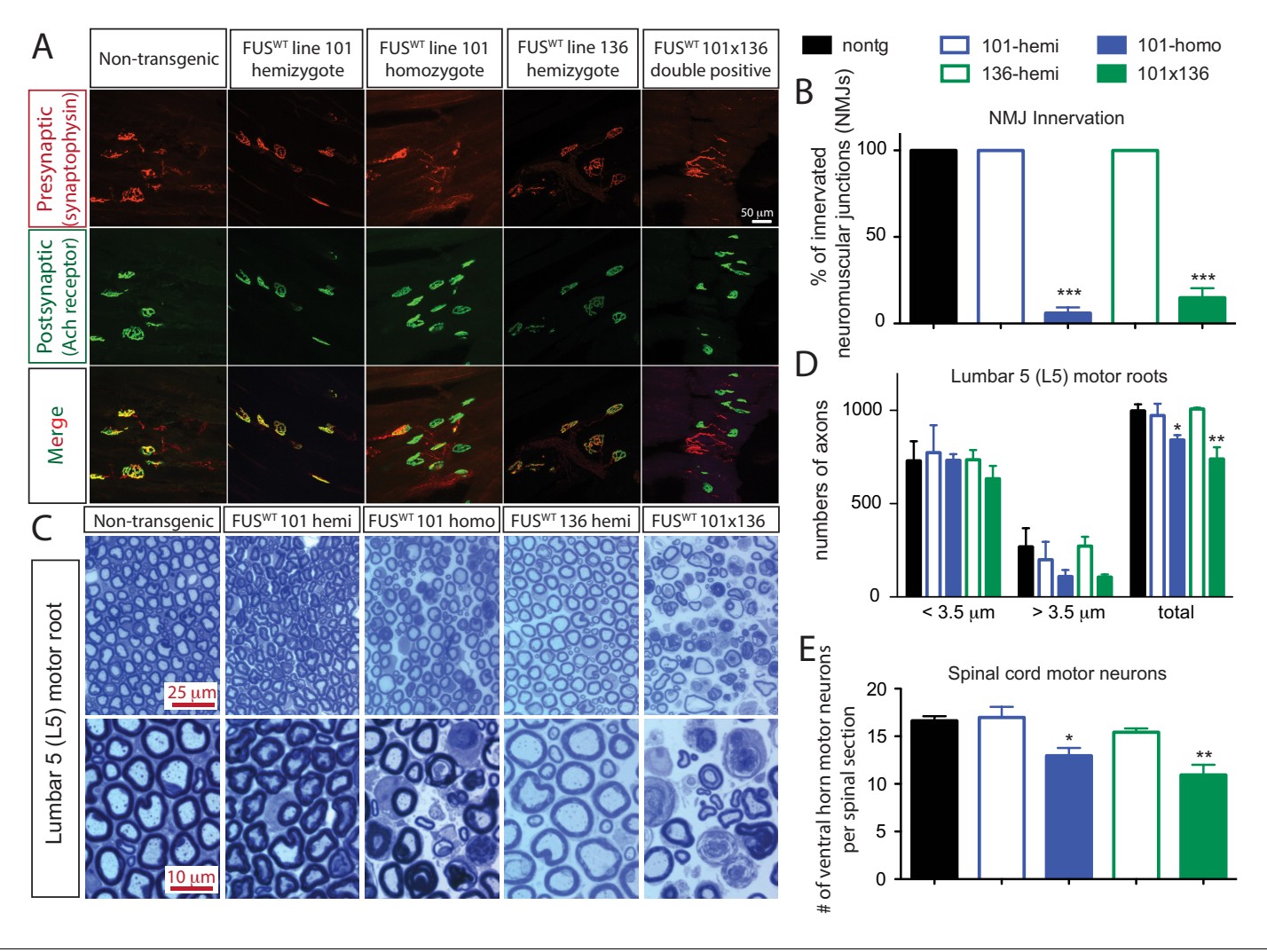

**Figure 4.** Acute sensitivity to FUS level in lower motor neuron circuit. (A) These representative images of the fluorescent staining of neuromuscular junctions (NMJs) in the gastrocnemius of 30-day-old animals with anti-synaptophysin (red), Fluoromyelin red (red) and α-Bungarotoxin Alexa 488 (green). (B) Quantification of NMJ innervation. The data represent the average of at least three animals ± SEM. (***: p<0.001, **: p<0.01, *: p<0.05). (C) Toluidine blue staining of lumbar 5 (L5) motor roots in 30-day-old animals. (D) Quantification of L5 motor roots in illustrated in (C). The data represent the average of at least three animals ± SEM. (**: p<0.01, *: p<0.05). (E) Quantification of ventral ChAT positive motor neurons in spinal cords of 30-day-old animals. The data represent the average of at least three animals ± SEM. (**: p<0.01, *: p<0.05).

DOI: https://doi.org/10.7554/eLife.40811.014

The following source data and figure supplement are available for figure 4:

**Source data 1.** Lower motor neuron system in FUS over-expressing mice.
DOI: https://doi.org/10.7554/eLife.40811.015

**Figure supplement 1.** Sensory root degeneration in *prnp*-FUS mice.
DOI: https://doi.org/10.7554/eLife.40811.016

## Altered RNA processing functions with elevated FUS expression

To identify how increased levels of FUS affect RNA maturation or stability, we performed transcriptomic analysis using total RNA isolated from 30-day-old spinal cords of non-transgenic mice as well as mice hemizygous and homozygous for the FUS^WT transgene (n = 3 per condition) at the 30 day old time point. Although RNAs in non-transgenic and hemizygous FUS^WT mice were almost indistinguishable at this age (only 7 RNA changes in comparing heterozygous FUS^WT and non-transgenic mice) (*Figure 6—figure supplement 1*), principal component analysis (PCA) and unsupervised

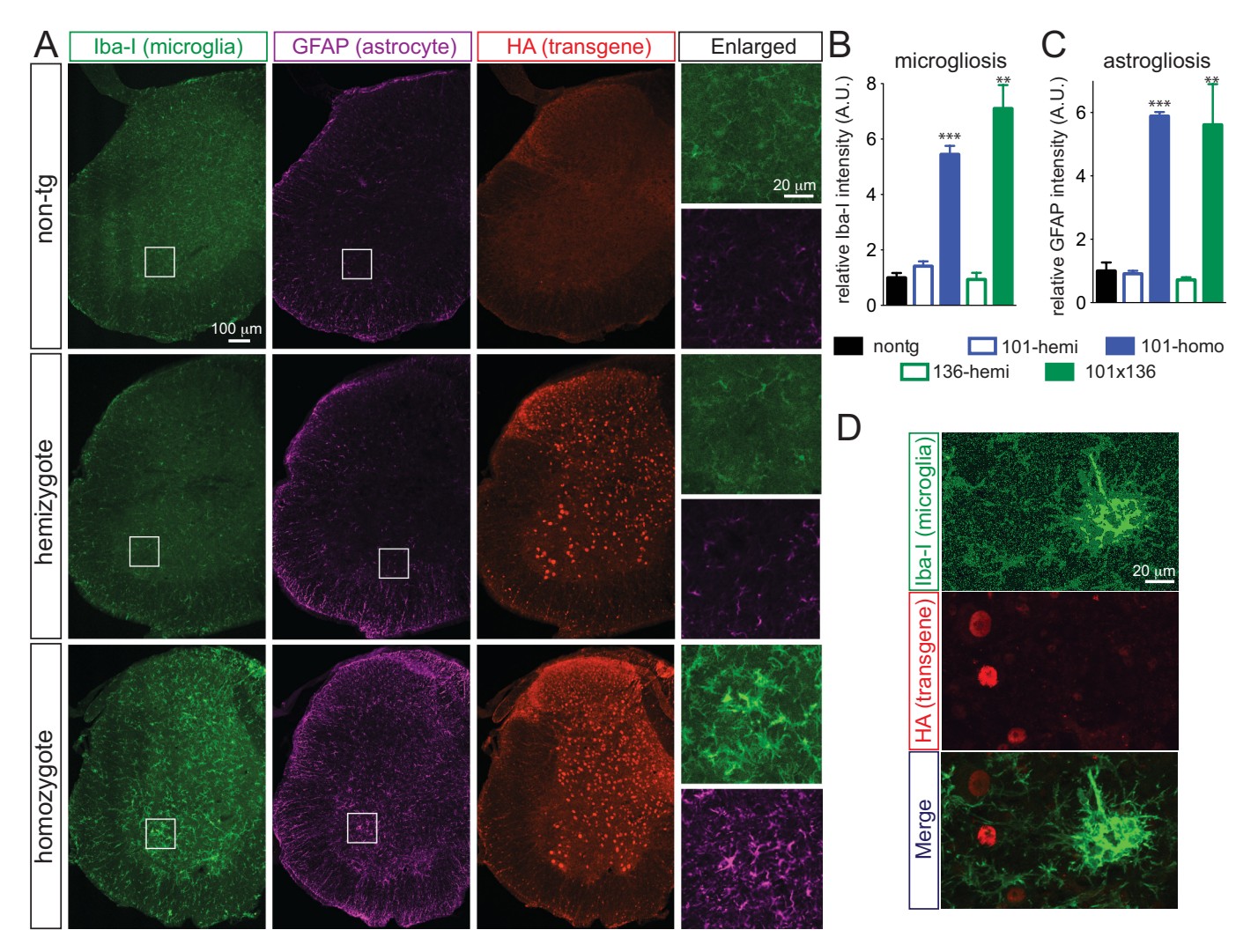

**Figure 5.** Prominent glial activation in *prnp*-FUS transgenic mice. (**A**) Representative images of lumbar spinal cord sections from non-transgenic, FUS^WT hemizygote and homozygote transgenic animals stained with Iba-I for microglia (green), GFAP for astrocyte (magenta) and HA for the transgene (red). (**B**) and (**C**) are the quantification of relative intensity of Iba-I and GFAP immunofluorescence signals. Signals of homozygote mice were set at 100%. (**D**) Double labeling of transgene FUS^WT (HA) and microglia (Iba-I). (**D**) Double labeling of transgene (HA) and microglia (Iba-I) in the spinal cords of homozygote FUS^WT transgenic mice showed that microglia do not express FUS.

DOI: https://doi.org/10.7554/eLife.40811.017

The following source data and figure supplement are available for figure 5:

**Source data 1.** Quantification of microgliosis and astrogliosis.
DOI: https://doi.org/10.7554/eLife.40811.018
**Figure supplement 1.** Elevated astrogliosis and microgliosis in *prnp*-FUS mice.
DOI: https://doi.org/10.7554/eLife.40811.019

hierarchical clustering revealed that homozygous FUS^WT animals had a highly distinct RNA profile (*Figure 6A–B*). Comparison of RNAs from 30 day-old non-transgenic and homozygous FUS^WT mice revealed 4081 expression changes (FDR corrected $p<0.1$, effect size $> \sqrt{2}$), with 2228 up- and 1853 down-regulated genes. Based on the known FUS-binding sites (*Lagier-Tourenne et al., 2012*), 15.1% of up-regulated genes (337 out of 2228 genes) and 26.1% of down-regulated genes (483 out of 1853 genes) are bound by FUS. Unbiased gene ontology (GO) biological process analyses of these differentially expressed genes (DEGs) showed distinct themes of up and down-regulated genes. The top up-regulated GO categories are involved in defense response, innate immune

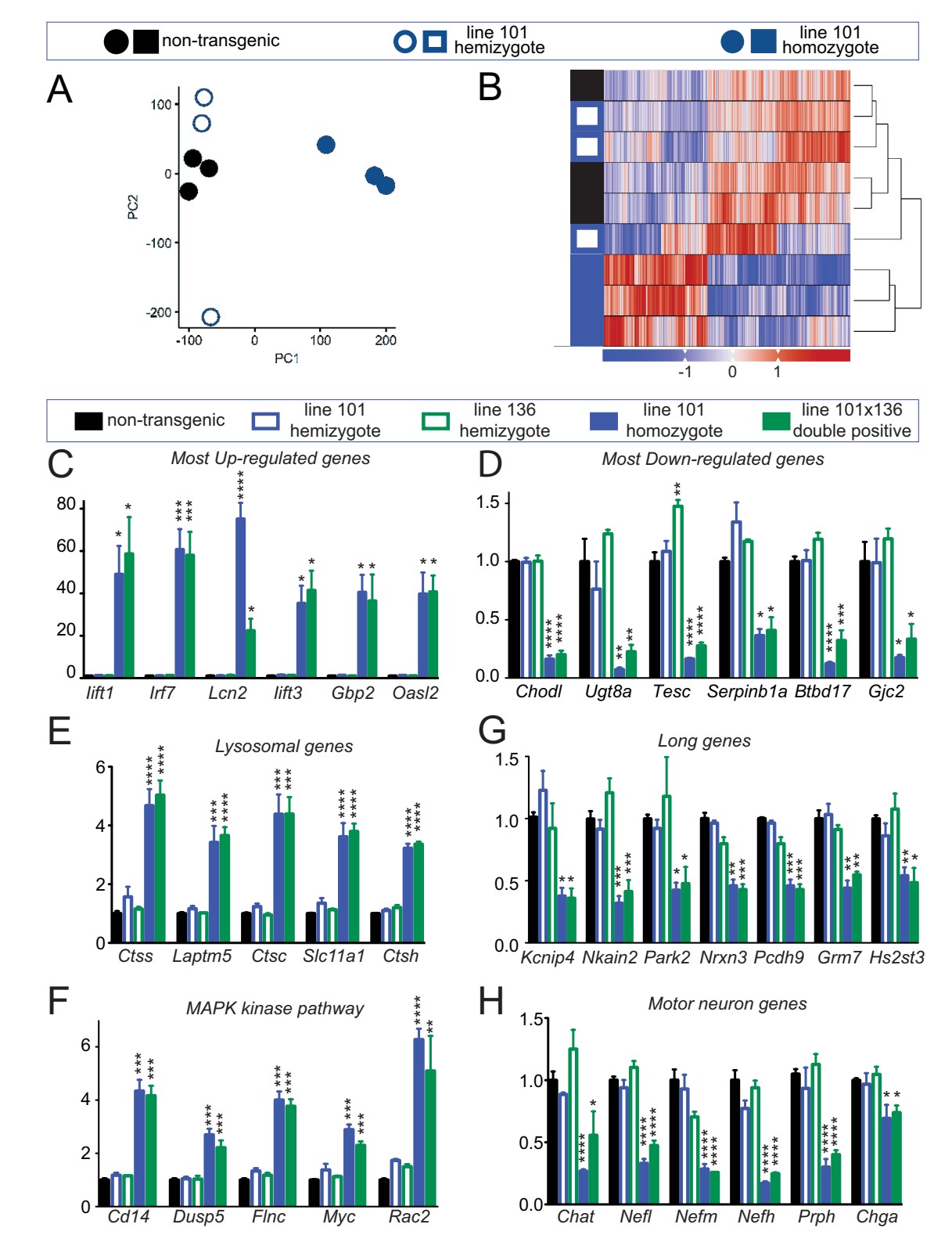

**Figure 6.** Altered RNA processing function in mice overexpression FUS. (**A**) Principal Component Analysis (PCA) of differentially expressed genes in spinal cords from 30-day-old non-transgenic, hemizygous and homozygous FUS[WT] mice. PCA of count data across all three conditions shows a clear separation of homozygous FUS mice separately from both non-transgenic and hemizygous FUS mice across the first two principal components. (**B**) Hierarchical clustering of gene-centered count data cleanly categorizes the data, clustering homozygous FUS mice separately from both non-transgenic

*Figure 6 continued on next page*

Figure 6 continued

and hemizygous FUS mice. (C–H) qRT-PCR validation of selective genes identified by RNA-seq: most up-regulated genes (C), most down-regulated genes (D), genes involved in lysosome function (E), genes involved in MAPK kinase pathway (F), genes with exceptional long introns (G), motor neurons genes (H). The data represent the average of at least three animals per genotype ±SEM. The changes are specific to doubly transgenics and transgene homozygotes, but not other genotypes.

DOI: https://doi.org/10.7554/eLife.40811.020

The following source data and figure supplements are available for figure 6:

Source data 1. qRT-PCR validation for differentially expressed genes (DEGs).
DOI: https://doi.org/10.7554/eLife.40811.021
Figure supplement 1. Distinct expression profiles in mice overexpression FUS.
DOI: https://doi.org/10.7554/eLife.40811.022
Figure supplement 2. Gene ontology enrichment (Biological process) of differentially expressed genes in FUS overexpressing mice.
DOI: https://doi.org/10.7554/eLife.40811.023
Figure supplement 3. No apparent gene expression changes in pre-symptomatic FUS over-expressing mice.
DOI: https://doi.org/10.7554/eLife.40811.024

response and regulation of immune system process, whereas top down-regulated GO categories are primarily metabolic, including steroid biosynthesis, cellular nitrogen compound metabolism process and cholesterol biosynthesis (*Figure 6—figure supplement 1*, *Supplementary file 1a-1b*). To better understand the biological processes and pathways affected by increased FUS level, these DEGs were further classified using KEGG (Kyoto Encyclopedia of Genes and Genomes) database. This analysis showed that the KEGG pathways of the most up-regulated genes are involved in lysosomes, antigen processing and presentation, cytokine-cytokine receptor interactions, the MAPK or p53 signaling pathways, and ECM-receptor interactions (*Supplementary file 1c*), whereas KEGG pathways of the most down-regulated genes were involved in steroid biosynthesis and terpenoid backbone biosynthesis (*Supplementary file 1d*). The top six most up-regulated RNAs (*Ifit1*, *Irf7*, *Lcn2*, *Ifit3*, *Gbp2*, and *Oasl2*) with a > 20 fold change and the six most down-regulated RNAs (*Chodl*, *Ugt8a*, *Tesc*, *Serpinb1a*, *Btdb17* and *Gjc2*) were validated by RT-qPCR in both doubly hemizygous and homozygous (line 101) FUS^WT spinal cords (*Figure 6C,D*). Similarly, up-regulated RNA levels of lysosomal genes (including *Ctss*, *Lamptm5*, *Ctsc*, *Slc11a1*, and *Ctsh* - *Figure 6E*) and genes involved in the MAPK kinase pathway (*Cd14*, *Dusp5*, *Flnc*, *Myc* and *Rac2* - *Figure 6F*) were also confirmed by qRT-PCR in both doubly hemizygous and homozygous (line 101) FUS^WT mice.

We have previously shown that genes with exceptionally long introns (>100 kb per average intron length), which encode proteins that are essential for neuronal functional and integrity, depend on both TDP-43 and FUS, with reduction in either protein reducing accumulation of the corresponding RNAs (*Lagier-Tourenne et al., 2012*; *Polymenidou et al., 2011*). Accumulation of the RNAs from these very long genes (including *Kcnip4*, *Nkain2*, *Park2*, *Nrxn3*, *Pcdh9*, *Grm* and *Hs2st3*) was also reduced in both doubly hemizygous (line 101 and line 136) and homozygous (line 101) FUS^WT mice (*Figure 6E*). Additionally, all four genes encoding subunits of neuronal intermediate filaments (*Nefl*, *Nefm*, *Nefh*, and *Prph*), together with the motor neuron-specific gene *Chat*, were down-regulated in spinal cords of doubly hemizygous and homozygous (line 101) FUS^WT mice, compared to age-matched non-transgenic and either hemizygous mouse lines (*Figure 6F*). Taken together, the data suggest that neurons are more susceptible to the increasing wild type human FUS levels, and increasing FUS levels by as little as 2-fold produces a counterintuitive, apparent loss of FUS function for maturation of the pre-mRNAs encoded by genes with exceptionally long introns.

Of note, the majority of mRNAs showed comparable expression levels with those of control mice, when qRT-PCR were performed for the P14 mice (*Figure 6—figure supplement 3*). At this age, the mice overexpressing FUS showed no apparent phenotype (*Figure 3*), suggesting that the massive transcription changes at late stage are dominated by the effects of neurodegeneration. Therefore, FUS over-expression does not induce widespread transcription changes at P14, but does lead to toxicity, and at late stages, this involves alterations in RNA metabolism (*Figure 6E*) and autophagy (see below).

# Transcriptome perturbations from FUS overexpression are distinct from endogenous FUS knockdown

Because of the autoregulatory mechanism for FUS expression, the drastic expression changes observed in the both doubly hemizygous and homozygous (line 101) FUS$^{WT}$ mice may be due to the reduction of endogenous mouse FUS protein, or the over-expression of functional human wild-type FUS protein, or both (*Figures 1*, *3* and *7A*). To determine whether the expression changes were primarily caused by the loss of endogenous FUS function and/or the dose-increased levels of FUS leading to gain-of-toxicity(ies), we compared the transcriptomic changes found in our 30 day old homozygous overexpressing FUS$^{WT}$ mice (OE) with those reported in mice with reduced FUS levels upon anti-sense oligonucleotide (ASO)-mediated knock-down (KD) (*Lagier-Tourenne et al., 2012*). In addition to the qualitative separation and clustering of the various groups observed in the PCA and clustered heatmaps (*Figure 7B–D*), we tested the divergence of transcriptome changes from wild type in both overexpression and knockdown models. We make the following assumption: to the extent that the observed phenotype in the overexpression samples stems primarily from the loss of endogenous FUS, transcriptome changes in both models should be reflected in a shared set of genes, with shared direction of expression changes. Conversely, if the observed overexpression phenotype was the result of FUS overexpression, genes perturbed in both conditions would be expected to display expression changes in opposite directions.

Differentially expressed genes are thus classified into two main categories: Similarly perturbed, in which the directions of expression change are the same in both conditions, and conversely perturbed, in which expression changes are in opposing directions. The bias of the proportional distribution of similarly and conversely perturbed genes is then used to identify the primary factor. If the distribution is opposite-dominant, with more conversely perturbed genes, FUS overexpression is likely the factor of interest, that is, the over-expression of FUS could drive the observed phenotype. If the distribution is instead similar-dominant, we infer that FUS reduction is the key factor. A schematic overview is shown in *Figure 7—figure supplement 1A*.

To determine the significance of this proportional shift, we can perform a binomial test of the proportion against the null model. The null model represents a case in which the genes are not differentially expressed. As such, we expect gene fold changes to be normally distributed around a mean of 0. Thus, P(conversely perturbed)=P(similarly perturbed)=0.5, resulting in a 50–50 distribution of genes in each category. Genes passing our significance cutoff (FDR < 0.1) demonstrated a marked opposite-dominant pattern, with a conversely perturbed proportion of 0.629 ± 0.017 (95% confidence interval) (*Figure 7D*). In contrast, proportional analysis on genes above our significance cutoff showed proportions close to the Null Model, with a conversely perturbed proportion of 0.533 ± 0.014 (95% Confidence interval) (*Figure 7—figure supplement 1B*). While this 'insignificant' gene set remained conversely perturbed, we note that the FDR cutoff is an arbitrary value, and that the gene set still contains genes with low p-values. This is clearly shown when the analysis is repeated across various cutoff values, in which the proportion of the 'insignificant' set approaches the Null Model (*Figure 7—figure supplement 1C*). Taken together, the lower motor neuron disease associated with dose-dependent increased accumulation of human FUS$^{WT}$, is primarily driven by gain-of-toxicity rather than loss of function of endogenous FUS.

Based on the known FUS-binding sites (*Lagier-Tourenne et al., 2012*), 23% of oppositely-regulated mRNAs (conversely perturbed category: expression changes are in opposing directions in FUS over-expression and knockdown conditions, 199 out of 862 genes) and 31% of co-regulated mRNAs (similarly perturbed category: the directions of expression change are the same in both FUS overexpression and knockdown conditions, 158 out of 507 genes) are bound by FUS. In the conversely perturbed category, FUS binds to 27% and 17% of mRNAs that are either down- or up-regulated in FUS-overexpression conditions, whereas FUS binds to 31% of the down-regulated mRNAs (69 out of 222) and 31% up-regulated mRNAs (89 out of 285) in the similarly perturbed category (*Figure 7D*).

Intriguingly, unbiased gene ontology (GO) analyses of these four categories of DEGs revealed that distinct biological processes are affected in each group. In the category i OE-down/KD-up group, mitochondrial respiratory chain and ribose phosphate metabolic process are affected; in the category ii OE-down/KD-up group, stem cell population maintenance and cytokine-mediated signaling pathway; in the category iii OE down/KD down group, central nervous system myelination, and regulation of synaptic plasticity are affected; in the category iv OE up/KD up group, innate immune

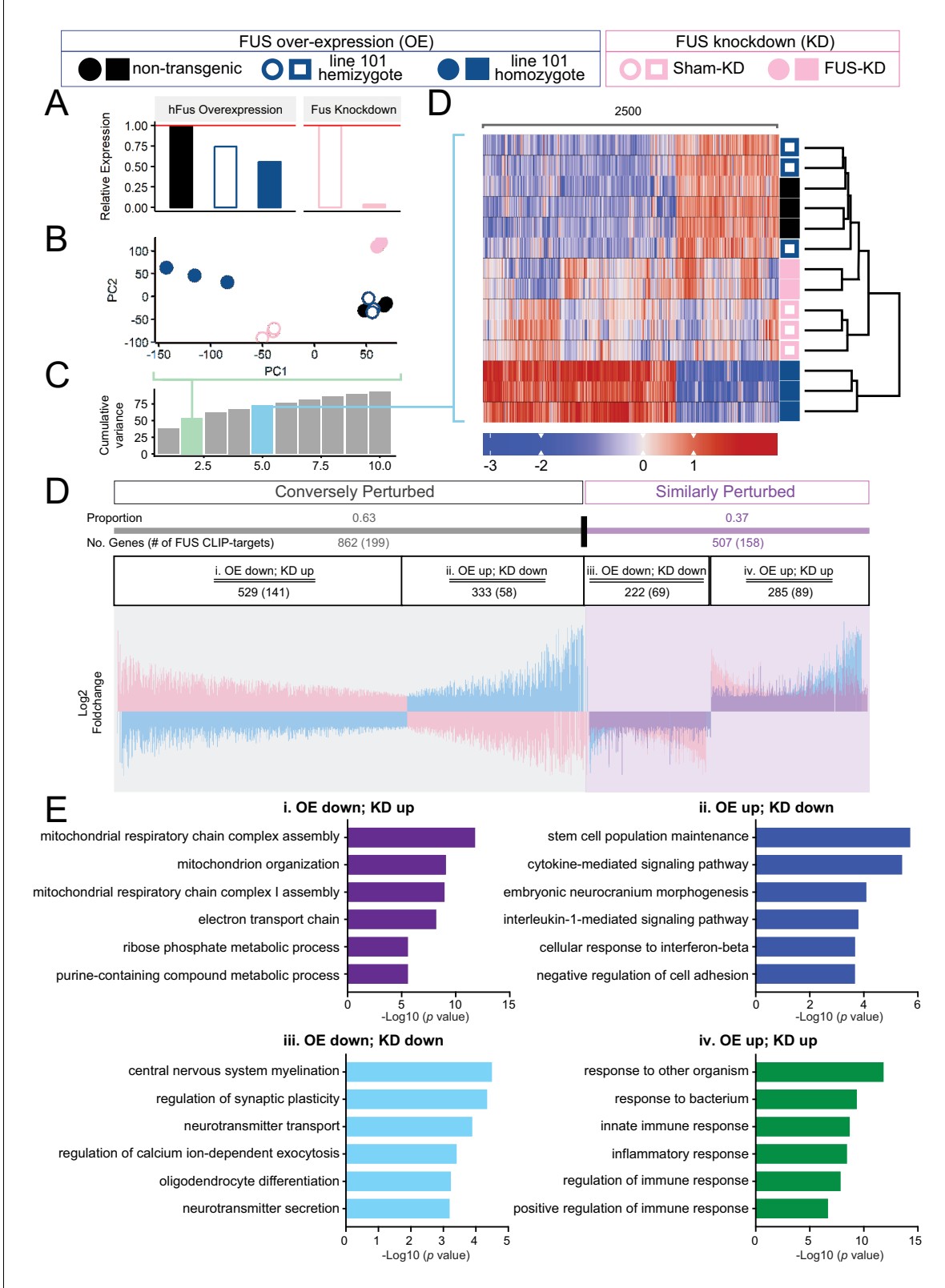

**Figure 7.** Distinct expression profiles between FUS over-expression and knockdown within the CNS of mice. (**A**) FUS overexpression (OE) in the CNS of mice results in a dose-dependent reduction in endogenous FUS. The reduction was noted to be of a lower extent when compared to samples with antisense-oligonucleotide (ASO) knockdown (KD) against mouse FUS. The relative RNA expression data were based on RNA-seq results. (**B**) Batch-corrected Principal Components (PCs) Analysis. (**C**) Cumulative explained variance across the first 10 principal components. The two PCs for the PCA

*Figure 7 continued on next page*

*Figure 7 continued*

and the five PCs for the heatmap are highlighted in green and blue, respectively. (**D**) Biclustered heatmap of log-scaled gene TPM values shows a clear separation of the OE and KD samples. (**E**) Proportional expression plot of genes below the 0.1 FDR cutoff shows a clear opposite dominant pattern with an OE proportion of 0.629 ± 0.017 (95% confidence interval). Blue, pink and purple represent the line-101 homozygote, FUS-ASO-KD, and similar expression overlap regions respectively. (**F**) Gene ontology enrichment (biological process) of DEGs in FUS overexpressing (OE) and knockdown (KD) conditions. Total of four categories of gene expression patterns are classified: (i) OE-down/KD-up, (ii) OE-down/KD-up, (iii) OE down/KD down, and (iv) OE up/KD up. Enriched GO terms are: (i) in the OE-down/KD-up group, mitochondrial respiratory chain and ribose phosphate metabolic process are affected; (ii) in the OE-down/KD-up group, stem cell population maintenance and cytokine-mediated signaling pathway are affected; (iii) in the OE down/KD down group, central nervous system myelination, regulation of synaptic plasticity are affected; and (iv) in the OE up/KD up, innate immune response and inflammatory response are affected. The X-axis represents the log-scaled FDR-corrected p-value.

DOI: https://doi.org/10.7554/eLife.40811.025

The following figure supplement is available for figure 7:

**Figure supplement 1.** Transcriptomic perturbations from FUS over-expression are distinct from endogenous FUS knockdown.

DOI: https://doi.org/10.7554/eLife.40811.026

response and inflammatory response are affected (*Figure 7E*). The genes in each group are summarized in *Supplementary file 2a-d*. These findings suggest that (1) similar biological process, such as synaptic functions and immune response, can be affected in the similar fashion by FUS-OE and FUS-KD, and conversely, (2) FUS-OE and FUS-KD can have opposite effects on other biological pathways, such as mitochondrial functions.

## Rescue of FUS-null lethality by CNS-expression of wild type human FUS

The apparent loss of RNA processing function despite increased overall FUS levels from expression of human wild type or mutant FUS (and autoregulated reduction in mouse FUS) raised the possibility that the human FUS cannot functionally replace mouse FUS, even though there is a 95.1% homology (at protein level) between the two proteins. To rule out this possibility, we produced FUS$^{WT}$ mice (in a pure C57BL/6 background) in which both endogenous FUS alleles were disrupted (*Hicks et al., 2000*) (*Figure 8A*). While endogenous FUS-null mice in this background die shortly after birth, the FUS$^{WT}$ transgene complemented the essential FUS functions throughout early development albeit with lower Mendelian ratio (*Figure 8A–B*). Levels of RNAs from the long intron-containing genes, as well as the FUS-dependent up-regulated genes, lysosomal genes and MAPK pathway genes were assayed by qRT-PCR in RNAs extracted from spinal cords of aged-matched (40 day-old) *prnp*-FUS$^{WT}$::*Fus*$^{-/-}$ and *Fus*$^{+/+}$ mice (*Figure 8C–D*). Most were nearly fully restored to their levels in non-transgenic mice, confirming the functional rescue of FUS function, albeit the most-down regulated genes were only partially rescued (*Figure 8C*), suggesting that maturation of these RNAs is very sensitive to FUS level. Altogether the near complete rescue of RNA expression changes in the FUS$^{WT}$ transgenic mice with both endogenous mouse alleles disrupted indicates that wild type human FUS protein complements the loss of the endogenous mouse FUS protein. Taken together, these genetic and genomic data suggest that even a slight imbalance of FUS level is critical for its proper function in the CNS and increasing FUS level exerts gain-of-toxic effects.

## Protein homeostasis defects via inhibiting autophagy in mice expressing increased levels of wild type FUS

Upregulation of lysosomal genes in the homozygous FUS$^{WT}$ mice suggests changes in the status of protein homeostasis. We first examined whether the human transgenes form any aggregation. The majority of FUS was nuclear, with only a small portion in the cytosol in the neurons of homozygous FUS$^{WT}$ mice (*Figure 9A*). Although ubiquitin-positive aggregates were not observed (data not shown), accumulation of p62/SQSTM1 (sequestosome 1) accumulated found in the motor neurons of homozygotes FUS$^{WT}$ mice, but not the non-transgenic and hemizygote FUS$^{WT}$ mice nor neighboring non-motor neurons (*Figure 9A*), suggesting a potential autophagic deficit in these motor neurons.

To determine whether FUS regulates autophagy, we exploited a dual fluorescence mCherry-GFP-LC3 system, whose expression leads to formation of puncta upon its incorporation into autophagosomes (*Mizushima et al., 2010*) during nutrient deprivation media (NLM)-induced autophagy in neuronal like cells (Neuro2A) (*Young et al., 2009*). As GFP fluorescence is sensitive to pH, yellow puncta (red +green) are indicative of autophagosomes (neutral pH), while red puncta mark autophagosomes

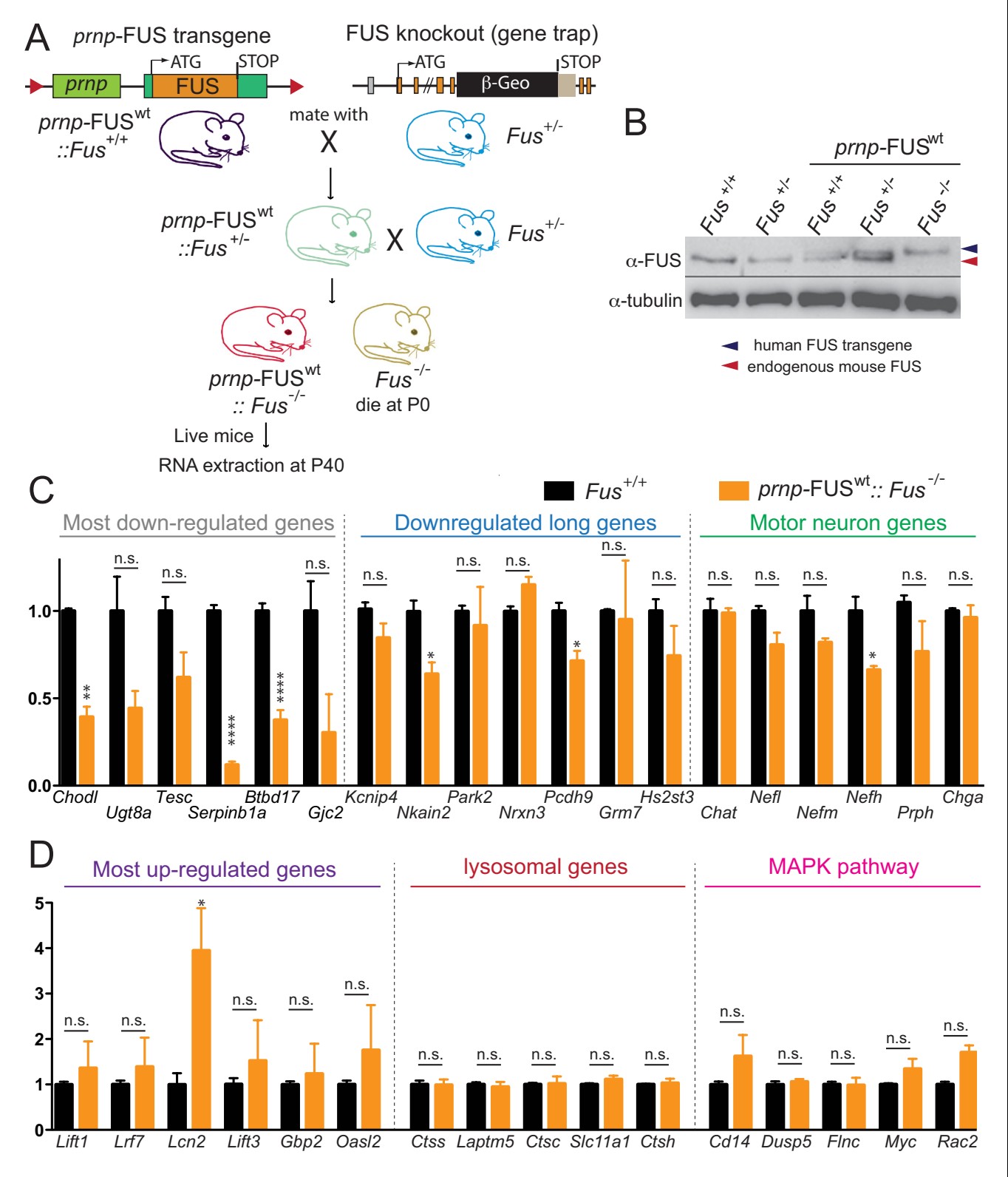

**Figure 8.** Rescue of FUS-null lethality by FUS expression in the CNS. (A) Breeding scheme to generate *prnp*-FUS[WT]:: Fus[-/-] mice. Fus[-/-] is generated through a gene-trap strategy. (B) Immunoblots of FUS in various genotypes: Fus[+/+], Fus[+/-], *prnp*-FUS[WT]:: Fus[+/+], *prnp*-FUS[WT]:: Fus[+/-], *prnp*-FUS[WT]:: Fus[-/-]. Blue and red arrowheads indicate human FUS transgene and endogenous FUS, respectively. (C–D) qRT-PCR analysis of validated genes in 40-day-old non-transgenic and *prnp*-FUS[WT]:: Fus[-/-] mice. The data represent the average of 3 animals ± SEM. (**: p<0.01, *: p<0.05). qRT-PCR analysis of
*Figure 8 continued on next page*

Figure 8 continued

validated genes in *Figure 6* showed that expression levels of genes with exceptionally long introns, motor neurons genes, lysosomal gens and MAPK pathway genes and the majority of most up-regulated genes could be rescued to the levels of non-transgenic animals. The majority of most down-regulated genes were not able to be rescued, suggesting that these genes may be extremely sensitive to FUS level.

DOI: https://doi.org/10.7554/eLife.40811.027

The following source data is available for figure 8:

**Source data 1.** qRT-PCR of DEGs in prnp-FUS mice under FUS-null background.

DOI: https://doi.org/10.7554/eLife.40811.028

that have fused with lysosomes to create autolysosomes (acidic pH) (*Figure 9B*). As expected, after 8 hr of NLM treatment of Neuro2A cells transiently expressing mCherry-GFP-LC3 and the blue fluorescence protein (BFP) as a control, robust autophagy (setting >7 puncta for autophagy activation) was induced as revealed by the accumulation of LC3 puncta in 45% of the cells (*Figure 9C,D*). However, increased expression of BFP-tagged wild-type FUS, FUS$^{R514C}$, or FUS$^{R521C}$ almost completely inhibited nutrient deprivation driven autophagic activation (*Figure 9E–F*), with a significant reduction in the fraction of red puncta that correspond to autolysosomes and an increase in the number of autophagosomes (yellow puncta) (*Figure 9C,E*). Immunofluorescence of p62 is used as an indirect measure of the autophagic flux (*Larsen et al., 2010*), as association of p62 with autophagosomes drives its degradation after fusion with autolysosomes. As anticipated, p62 puncta accumulation was reduced in Neuro2A cells expressing BFP upon NLM addition (*Figure 9C*, two right top and middle panels). However, increased levels of FUS$^{WT}$ did not suppress the NLM-mediated p62 puncta accumulation, but rather increased the percentage of cells with p62 puncta (*Figure 9C,F*), indicative of the compromised autophagic flux pathway. These findings indicate that increased expression of wild-type or either of these ALS-linked FUS mutants inhibits autophagy.

## Discussion

Here we report the generation and characterization of transgenic mice that express wild type or either of two ALS-linked mutants of human FUS (R514G and R521C) broadly in the CNS, with transgene expression levels and patterns to that of endogenous mouse FUS. These mice developed progressive and mutant-enhanced motor deficits accompanied by ALS-like lower motor neuron pathology. Furthermore, FUS autoregulates its own protein expression level in the CNS. An increase in expression of wild type FUS can saturate this autoregulation and thereby sharply accelerates disease phenotypes and triggers early mortality accompanied by disturbances in both protein homeostasis and RNA processing (*Figure 10*). In particular, our data reveal a role for FUS in regulating autophagy. Furthermore, increasing FUS levels could affect neuronal and synaptic functions by causing a loss of RNA processing for particularly long genes. Thus, disruption of FUS homeostasis incurs gain-of-function toxicity directed against autophagy and loss-of RNA processing function, thereby providing an explanation for how FUS can promote disease pathogenesis in the absence of mutation (*Figure 10*).

FUS auto-regulation is known to maintain normal FUS levels and the underlying mechanisms have been previously described in cultured cells (*Zhou et al., 2013*; *Dini Modigliani et al., 2014*) and in a recent humanized FUS mouse model (*López-Erauskin et al., 2018*). One striking feature of all 7 FUS transgenic mouse lines we produced here is FUS autoregulation in the adult mouse CNS, which leads to mice with transgene levels that are close to mouse FUS in non-transgenic animals While all transgenic animals (FUS$^{WT}$, FUS$^{R514G}$ and FUS$^{R521C}$) initially appeared normal, with typical development and weight gain into adult life, by 12 months of age wild-type and mutant FUS mice develop abnormal motor phenotypes, including clasping, lower posture and reduced hind limb spread, decreased stride-length, increased EMG activity, a reduction in the number of spinal cord motor neurons and their axons and a significant loss of hind limb neuromuscular innervation. In addition, FUS$^{WT}$ mice are less affected compared with FUS$^{R514G}$ mice, suggesting a mutant-enhanced toxicity. These abnormalities are all found in ALS pathogenesis in mutant SOD1 mouse models and patients (*Bruijn et al., 2004*). Despite continued mutant accumulation, however, disease is at best very slowly progressive and does not lead to lower limb paralysis by ages up to 24 months. Furthermore,

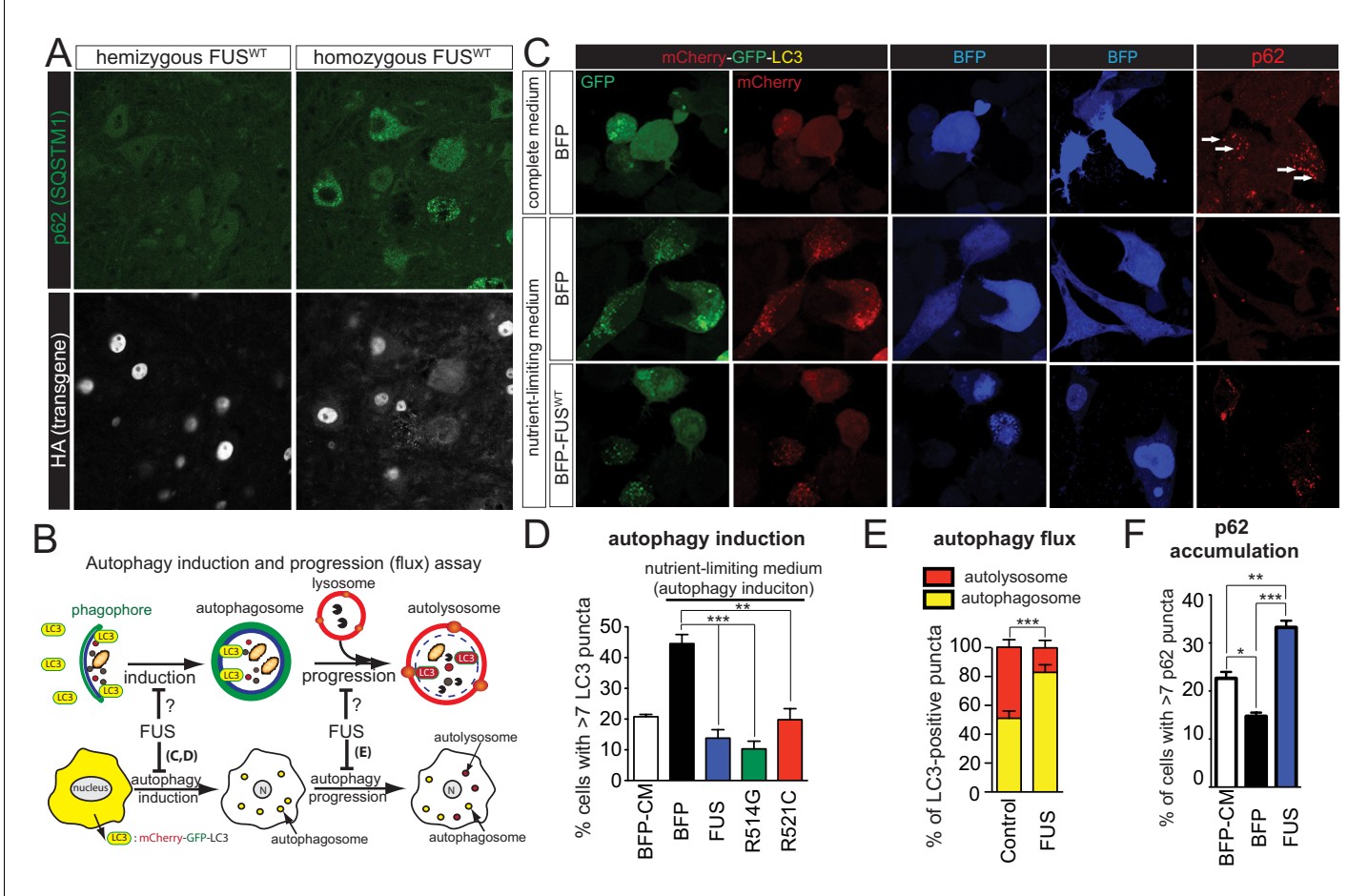

**Figure 9.** Gain of protein toxicity via autophagy inhibition in *prnp*-FUS transgenic mice. (**A**) Representative images of lumbar spinal cord sections from hemizygote and transgene homozygote FUS^WT transgenic animals stained with p62 (SQSTM1) (green) and HA for human FUS transgene (grey). Scale bar is 20 μm. Cytoplasmic accumulations of p62 were visible in the ventral horn motor neurons of the spinal cord in transgene homozygous mice, but not other genotypes. (**B**) Schematics of autophagy induction and progression (flux) assay. Double-tagged LC3 (mCherry-GFP-LC3) were used to visualize the induction and progression of autophagy. Upon autophagy induction, LC3 is post-translationally modified with a lipid group and localize to autophagosome. Autophagosomes progress to autolysosomes by fusing with lysosome. Fluorescent signal of GFP is pH-sensitive and is quenched in autolysosome. (**C**) Representative images of Neuro2A cells co-transfected with BFP (blue fluorescent protein) or BFP-tagged wild type FUS with mCherry-GFP-LC3 (left panel) or immuno-stained with p62 (right panel). Nutrient-limiting medium was used to induce autophagy. (**D**) Quantification of autophagy induction under nutrient-limiting medium conditions. Over-expression of wild type and ALS-linked mutants (R514G and R521C) in FUS inhibit autophagy induction. (**E**) Quantification of autophagy flux based on the numbers of autophagosome and autolysosome. Overexpression of FUS inhibits autophagy flux. (**F**) Quantification of p62 accumulation in cells transfected with BFP or BFP-tagged FUS.

DOI: https://doi.org/10.7554/eLife.40811.029

The following source data is available for figure 9:

**Source data 1.** Quantification of autophagy assays.

DOI: https://doi.org/10.7554/eLife.40811.030

although R521C mutation is one of the most frequent FUS mutations in ALS, expression of R521C in mice did not produce motor axon loss, but rather sensory axon loss, which differs from the FUS-related ALS in human. By contrast, wild-type and R514G transgene produce both the motor and sensory axon degeneration. Thus, it is possible that the observed sensory deficits may be caused by transgene expression driven by the murine prion promoter. However, GEM bodies in the spinal cord motor neurons of FUS^R521C mice were significantly reduced (*Sun et al., 2015*), an indication of reduced SMN (survival motor neuron) function. This is reminiscent of the notion that altered circuit function could underlie the neurodegeneration as in spinal muscular atrophy (SMA) (*Mentis et al., 2011*), in which dysfunction of cholinergic sensory neurons and interneurons could lead to

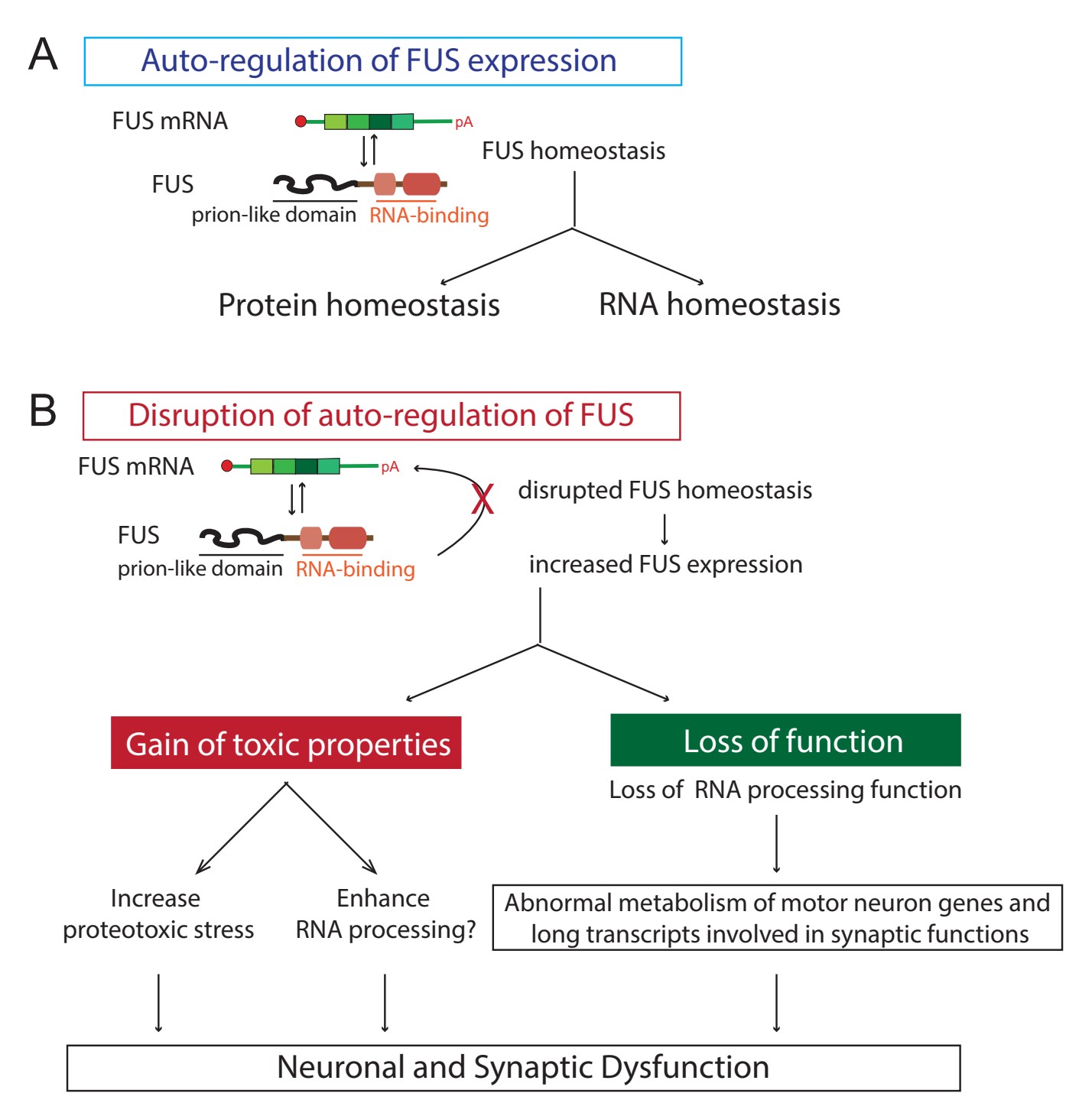

**Figure 10.** Proposed model of FUS-mediated neurodegeneration. FUS homeostasis is essential for maintaining both protein and RNA homeostasis. The level of FUS is maintained possibly through nonsense-mediated decay and/or miRNA-mediated mechanisms (see text for references). Elevated FUS level produces both gain-of-toxic properties by either increasing proteotoxic stress or expression of stress genes and loss-of RNA-processing function in which genes involved in long transcripts and synaptic functions are selectively affected. Together they cause neuronal and synaptic dysfunction and eventual neuronal death.

DOI: https://doi.org/10.7554/eLife.40811.031

motoneuron degeneration (*Imlach et al., 2012*; *Lotti et al., 2012*). Further studies are required to investigate the possibilities of altered motor-sensory circuit in FUS pathogenesis.

More importantly, we uncovered that the homeostatic levels of FUS are essential for proper neuron function, as a 2-fold increase in FUS levels accelerated the disease at least 5-fold (*Figure 3*). Elevated expression levels of genes encoding the pathological hallmarks are not without precedent in other neurodegenerative diseases, such as duplication of the APP gene (amyloid precursor protein, causal for Alzheimer's disease) (*Sleegers et al., 2006*), and locus triplication of α-synuclein (causal for Parkinson's disease) (*Singleton et al., 2003*). Furthermore, ALS-causing mutations in the 3'-UTR of FUS gene leading to elevated FUS levels have been identified (*Sabatelli et al., 2013*). Similar findings, that variants within the 3'-UTR of *TARDBP* gene can lead to increased TDP-43 expression, have also been reported for FTD (*Gitcho et al., 2009*). Collectively, the evidence suggests that beside the disease-linked mutations in the coding regions, the factors leading to elevated expression of these key disease genes could contribute to disease pathogenesis.

CNS-expression of wild type human FUS rescues the FUS-null lethality in mice, suggesting that human FUS protein can functionally replace the mouse FUS gene (*López-Erauskin et al., 2018*). Furthermore, genome-wide expression profiling reveals gene changes that are largely distinct with FUS-knockdown using antisense oligonucleotides. Indeed, work by Kino and colleagues showed that the behavior and pathological phenotypes caused by FUS deletion are largely distinct from those observed in ALS (*Kino et al., 2015*), thereby supporting the notion that it is unlikely that FUS-mediated ALS is initiated by the loss-of-FUS-function. Nevertheless, varying FUS levels, that is, FUS over-expression (OE) and FUS knockdown (KD) conditions, could affect distinct and similar biological processes. For example, similar biological processes, such as synaptic functions, myelination, and immune response, can be affected by either increased or lowered levels of FUS. Conversely, FUS-OE and FUS-KD may have opposite effects on mitochondrial function. Interestingly, dysfunctions of these processes, for example, synaptic deficits (*López-Erauskin et al., 2018*; *Sephton et al., 2014*) and mitochondria damage (*Deng et al., 2015*), have been proposed as potential FUS-mediated pathologies. As (i) FUS pathology is associated with loss of nuclear FUS and concomitant cytosolic FUS accumulation (*Neumann et al., 2009*; *Dormann et al., 2010*), and (ii) the strong phenotype produced by over-expression of wild type FUS did not correlate with FUS aggregation as the majority of FUS remains in the nucleus, a likely scenario is that the toxic cascade is initiated at the nucleus with dysregulation of RNA processing being one of the potential mechanisms, and subsequently, is exacerbated by the loss of nuclear FUS.

Autophagy dysfunction has been implicated in various neurodegenerative diseases, including ALS (*Wong and Cuervo, 2010*; *Nixon and Yang, 2012*). We showed here that increased FUS expression (wild type or disease-linked mutants) is sufficient to inhibit autophagy induction and flux using both LC3 and p62 as markers (*Figure 9*). Interestingly, the majority of FUS protein remained in the nucleus of motor neurons, but not in other cells, in the spinal cord and co-localized with p62 (*Figure 9*), suggesting perturbed protein homeostasis in motor neurons and that motor neurons may be more susceptible to elevated FUS levels. Furthermore, recent discoveries of many ALS-FTD genes, including *SQSTM1* (*Fecto et al., 2011*), *OPTN* (*Maruyama et al., 2010*; *Wong and Holzbaur, 2014*), *TBK1* (*Cirulli et al., 2015*; *Freischmidt et al., 2015*; *Pottier et al., 2015*), and *VCP* (*Johnson et al., 2010*), that are involved in autophagy, indicates that autophagy dysfunction plays a pivotal role in ALS-FTD pathogenesis. Further studies investigating the molecular mechanisms of how overexpression of FUS inhibits autophagy will be urgently needed.

Activation of astrocytes (astrogliosis) is one hallmark of ALS pathology (*Schiffer et al., 1996*). Reactive astrocytes become apparent in lumbar spinal cord sections of in all three genotypes in aged (10 ~ 12 month-old) animals. Similar to the endogenous protein, human FUS transgenes are expressed in both neurons and astrocytes. Since pathological aggregation of FUS can be found in neurons and non-neurons, including astrocytes (*Neumann et al., 2009*), the signal initiating astrogliosis could come either from motor neurons, from astrocytes, or from both. This is reminiscent of the non-cell autonomous toxicity mediated by mutant SOD1 in ALS pathogenesis (*Ilieva et al., 2009*) and can be tested in this set of transgenic mice by using tissue-specific Cre recombinase-mediated excision of the transgene in neurons or astrocytes (*Lobsiger et al., 2009*; *Yamanaka et al., 2008*; *Boillée et al., 2006*). However, in contrast to SOD1-mediated toxicity, it appears that neither endogenous nor transgenic FUS is expressed in microglia (*Figure 5*, see also

(*Qiu et al., 2014*)), suggesting that disease pathogenesis is not driven by FUS-mediated toxicity within the microglia.

In conclusion, our FUS mouse models resemble the endogenous FUS expression pattern and levels and developed age-dependent and mutant-enhanced phenotypes. We provide evidence that elevated FUS levels produce gain-of-toxic properties by overriding FUS autoregulation, disrupting protein and RNA homeostasis. These disruptions are, at least in part, due to autophagy inhibition causing proteotoxic stress and a loss of normal RNA processing function, in which genes with long transcripts and involved in synaptic function are selectively affected. Together, these abnormalities promote neural and synaptic dysfunction, culminating in neuron death. Thus, maintaining FUS homeostasis, which in turn is required for proper protein and RNA homeostasis, is essential for normal neuron function (*Figure 10*).

# Materials and methods

## Key resources table

| Reagent type (species) or recourse | Designation | Source or reference | Identifier | Additional information |
|---|---|---|---|---|
| Gene (*Homo sapines*) | FUS | NA | Gene ID: 2521 | |
| Strain, strain background (*Mus musculus*) | Mouse/B6N. Cg-Tg(Prnp-FUS-wt) line 101 | this study | murine prion promoter driven FUS cDNA (human wild type FUS) | C3H and C57BL/6 hybrid backcross to C57BL/6J for more than five generations |
| Strain, strain background (*Mus musculus*) | Mouse/B6N. Cg-Tg(Prnp-FUS-wt) line 136 | this study | murine prion promoter driven FUS cDNA (human wild type FUS) | C3H and C57BL/6 hybrid backcross to C57BL/6J for more than five generations |
| Strain, strain background (*Mus musculus*) | Mouse/B6N. Cg-Tg(Prnp-FUS-R514G) line 124 | this study | murine prion promoter driven FUS cDNA (human R514G FUS) | C3H and C57BL/6 hybrid backcross to C57BL/6J for more than five generations |
| Strain, strain background (*Mus musculus*) | Mouse/B6N. Cg-Tg(Prnp-FUS-R521C) line 135 | this study | murine prion promoter driven FUS cDNA (human R521C FUS) | C3H and C57BL/6 hybrid backcross to C57BL/6J for more than five generations |
| Strain, strain background (*Mus musculus*) | C57BL/6 mFUS-/- | *Hicks et al., 2000* | | C57BL/6J background |
| Cell line (Mus musculus) | Neuro2A | ATCC | Cat# CCL-131, RRID:CVCL_0470 | Free of mycoplasma contamination. Species confirmed and authenticated by the Cytochrome oxidase 1 (CO1) barcode assay. |
| Transfected construct | pEBFP2-Nuc | Addgene | 14893, RRID:Addgene_14893 | |
| Transfected construct (*Homo sapines*) | BFP-wt-FUS | this study | | |
| Transfected construct (*Homo sapines*) | BFP-R514G-FUS | this study | | |

*Continued on next page*

*Continued*

| Reagent type (species) or recourse | Designation | Source or reference | Identifier | Additional information |
|---|---|---|---|---|
| Transfected construct (*Homo sapines*) | BFP-R521C-FUS | this study | | |
| Transfected construct (*Homo sapines*) | pDest-mCherry-EGFP-hLC3b | Addgene | 22418, RRID:Addgene_22418 | |
| Antibody | FUS | Santa Cruz | clone 4H11, sc-47711, RRID:AB_2105208 | 1:500 (WB), 1:100 (IF) |
| Antibody | human FUS, #14080, affinity purified | this study, Lopez-Erauskin et al., Neuron 2018 | peptide sequences for immunization, CKKKGSYSQQPSYGGQQ | 0.1 ug/ml (WB), 1 ug/ml (IF) |
| Antibody | mouse FUS, #14082, affinity purified | this study, Lopez-Erauskin et al., Neuron 2018 | peptide sequences for immunization, CKKKGGYGQQSGYGGQQ | 0.1 ug/ml (WB), 1 ug/ml (IF) |
| Antibody | HA | Bethyl Laboratories, Inc. | A190-238A, RRID:AB_2631898 | 1:5000 (WB), 1:1000 (IF) |
| Antibody | HA | Covance | mms-101P, RRID:AB_2314672 | 1:1000 (WB), 1:500 (IF) |
| Antibody | NeuN | Merck Millipore | MAB377, RRID:AB_2298772 | 1:1000 (IF) |
| Antibody | NeuN, Alexa-488 conjugate | Merck Millipore | MAB377X, RRID:AB_2149209 | 1:1000 (IF) |
| Antibody | GFAP | Protien Tech | 16825–1-AP, RRID:AB_2109646 | 1:1000 (IF) |
| Antibody | APC/CC1 | Merck Millipore | OP80, RRID:AB_2057371 | 1:1000 (IF) |
| Antibody | ChAT | Merck Millipore | AB144P, RID:AB_2079751 | 1:100 (IF) |
| Antibody | INPP4B | Cell Signaling Technology | #4039, RRID:AB_2126015 | 1:1000 (WB) |
| Antibody | KHC | Abcam | ab62104, RRID:AB_2249625 | 1:1000 (WB) |
| Antibody | HSP-90 | Enzo Life Sciences | ADI-SPA-846-D, RRID:AB_2039287 | 1:1000 (WB) |
| Antibody | Iba-I | Wako | 019–19741, RRID:AB_839504 | 1:1000 (IF) |
| Antibody | TDP-43 | ProteinTech | 10782–2-AP, RRID:AB_615042 | 1:1000 (IF) |
| Antibody | p62/sequestosome | Enzo Life Sciences | BML-PW9860, RRID:AB_2196009 | 1:500 (IF) |

*Continued on next page*

*Continued*

| Reagent type (species) or recourse | Designation | Source or reference | Identifier | Additional information |
|---|---|---|---|---|
| Antibody | BFP | abcam | ab32791, RRID:AB_873781 | 1:400 (IF) |
| Antibody | synaptophysin | Thermo Fisher Scientific | PA1-1043, RRID:AB_2199026 | 1:50 (IF) |
| Antibody | Goat anti mouse alexa 350 | Thermo Fisher Scientific | A21049, RRID:AB_141456 | 1:100 (IF) |
| Antibody | Goat anti rabbit alexa 594 | Thermo Fisher Scientific | A11037, RRID:AB_2534095 | 1:100 (IF) |
| Antibody | Donkey anti Rabbit FITC | Jacksons Immunol | 711-095-152, RRID:AB_2315776 | 1:500 (IF) |
| Antibody | Donkey anti Rabbit Cy3 | Jacksons Immunol | 711-165-152, RRID:AB_2307443 | 1:500 (IF) |
| Antibody | Donkey anti Rabbit Cy5 | Jacksons Immunol | 711-175-152, RRID:AB_2340607 | 1:500 (IF) |
| Antibody | Donkey anti Mouse FITC | Jacksons Immunol | 715-095-151, RRID:AB_2335588 | 1:500 (IF) |
| Antibody | Donkey anti Mouse Cy3 | Jacksons Immunol | 715-165-151, RRID:AB_2315777 | 1:500 (IF) |
| Antibody | Donkey anti Mouse Cy5 | Jacksons Immunol | 715-175-151, RRID:AB_2619678 | 1:500 (IF) |
| Antibody | Donkey anti Goat FITC | Jacksons Immunol | 705-095-147, RRID:AB_2340401 | 1:500 (IF) |
| Antibody | Donkey anti Goat Cy3 | Jacksons Immunol | 705-165-147, RRID:AB_2307351 | 1:500 (IF) |
| Antibody | Donkey anti Goat Cy5 | Jacksons Immunol | 705-175-147, RRID:AB_2340415 | 1:500 (IF) |
| Recombinant DNA reagent | pcDNA5-FRT-TO-GFP-wt-FUS | *Sun et al., 2015* | | |
| Recombinant DNA reagent | pcDNA5-FRT-TO-GFP-R514G-FUS | *Sun et al., 2015* | | |
| Recombinant DNA reagent | pcDNA5-FRT-TO-GFP-R521C-FUS | *Sun et al., 2015* | | |
| Sequence-based reagent | qRT-PCR primers, mouse *Chodl* | IDT | forward, 5'-CCAGATGTTGCATAAAAGTAAAGGA-3', reverse, 5'-TCCAGAACAATGCCAGTTCA-3' | |
| Sequence-based reagent | qRT-PCR primers, mouse *Ugt8a* | IDT | forward, 5'-CGAAGGACGCGCTATGAAG-3', reverse, 5'-CAAGGCCGATGCTAGTGTCT-3' | |

*Continued on next page*

*Continued*

| Reagent type (species) or recourse | Designation | Source or reference | Identifier | Additional information |
|---|---|---|---|---|
| Sequence-based reagent | qRT-PCR primers, mouse *Tesc* | IDT | forward, 5'-TTGAAAAGGAGTCGGCTCGG-3', reverse, 5'-CACCTGGTCCGGTTCCATC-3' | |
| Sequence-based reagent | qRT-PCR primers, mouse *Serpinb1a* | IDT | forward, 5'-TGACTTTTGGCAT GGGTATGTC-3', reverse, 5'-GTCATGCAAAAGCCGAGGAG-3' | |
| Sequence-based reagent | qRT-PCR primers, mouse *Btbd17* | IDT | forward, 5'-GGGACTGTGCTGCTGTCTTT-3', reverse, 5'-CTCACCACAGTAC AAATACCTGATG-3' | |
| Sequence-based reagent | qRT-PCR primers, mouse *Gjc2* | IDT | forward, 5'-GCCTGGAGAAGGTCCCAC-3', reverse, 5'-GTCAGCACAATGCGGAAGAC-3' | |
| Sequence-based reagent | qRT-PCR primers, mouse *Kcnip4* | IDT | forward, 5'-TTCATTGAAAGT TGCCAAAAA-3', reverse, 5'-CTACAAGTGGG GGCTTCAAC-3' | |
| Sequence-based reagent | qRT-PCR primers, mouse *Nkain2* | IDT | forward, 5'-GGGCTTCATCT ATGCCTGTT-3', reverse, 5'-GATGTCTTCTG AGGCCCTTG-3' | |
| Sequence-based reagent | qRT-PCR primers, mouse *Park2* | IDT | forward, 5'-CAGACAAGGAC ACGTCGGTA-3', reverse, 5'-GGGATCCCAG GAAGTCTTGT-3' | |
| Sequence-based reagent | qRT-PCR primers, mouse *Nrxn3* | IDT | forward, 5'-TTTCACCTGTG ACTGCTCCA-3', reverse, 5'- TTGCTGGCCAG GTATAGAGG-3' | |
| Sequence-based reagent | qRT-PCR primers, mouse *Pcdh9* | IDT | forward, 5'-GACAAGAGGAC CGAAGCAGA-3', reverse, 5'-GGTGTTGGTAT GGACCCAAG-3' | |
| Sequence-based reagent | qRT-PCR primers, mouse *Grm7* | IDT | forward, 5'-GACTCGGGGTG TACCAGAGA-3', reverse, 5'-TGGAGATTGTA AGCGTGGTG-3' | |
| Sequence-based reagent | qRT-PCR primers, mouse *Hs2st3* | IDT | forward, 5'-GGACGAGGACT GGACTGGTA-3', reverse, 5'-GGGCTTCTTGA GTGACGAAA-3' | |

*Continued on next page*

*Continued*

| Reagent type (species) or recourse | Designation | Source or reference | Identifier | Additional information |
|---|---|---|---|---|
| Sequence-based reagent | qRT-PCR primers, mouse *Chat* | IDT | forward, 5'-TCCGCTTCCGA GATGTTTCC-3', reverse, 5'-AACATAGGGCC GGTTCCTTC-3' | |
| Sequence-based reagent | qRT-PCR primers, mouse *Nefl* | IDT | forward, 5'-TGAGCCCTATTCC CAACTATTCC-3', reverse, 5'-GGTTGACCTGAT TGGGGAGA-3' | |
| Sequence-based reagent | qRT-PCR primers, mouse *Nefm* | IDT | forward, 5'-CCATCCAGCAGT TGGAAAAT-3', reverse, 5'-CGGTGATGCTT CCTGAAAAT-3' | |
| Sequence-based reagent | qRT-PCR primers, mouse *Mefh* | IDT | forward, 5'-CAGCTGGACAGT GAGCTGAG-3', reverse, 5'-CAAAGCCAATCCG ACACTCT-3' | |
| Sequence-based reagent | qRT-PCR primers, mouse *Prph* | IDT | forward, 5'-TGTGCCATTGTC AGGAGTCAG-3', reverse, 5'-CTGTCTGGTGTT CCTCTCTGG-3' | |
| Sequence-based reagent | qRT-PCR primers, mouse *Chga* | IDT | forward, 5'-GGTGCTGGACTT GGGATAGG-3', reverse, 5'-CAGAGACAATGC CCCCACTC-3' | |
| Sequence-based reagent | qRT-PCR primers, mouse *Iift1* | IDT | forward, 5'-GCATCACCTTC CTCTGGCTAC-3', reverse, 5'-GAATGGCCTGTTG TGCCAAT-3' | |
| Sequence-based reagent | qRT-PCR primers, mouse *Irf7* | IDT | forward, 5'-ACCCAAGGGG CCTTATTTGC-3', reverse, 5'-TCTACACAGGCA GTCTGGGA-3' | |
| Sequence-based reagent | qRT-PCR primers, mouse *Lcn2* | IDT | forward, 5'-AGCCACCATAC CAAGGAGCA-3', reverse, 5'-GGGGAGTGCTG GCCAAATA-3' | |
| Sequence-based reagent | qRT-PCR primers, mouse *Iift3* | IDT | forward, 5'-TGAGGACAACC GGAAGTGTG-3', reverse, 5'-TTTTCAGCACATT CTCCCCA-3' | |

*Continued*

| Reagent type (species) or recourse | Designation | Source or reference | Identifier | Additional information |
|---|---|---|---|---|
| Sequence-based reagent | qRT-PCR primers, mouse *Gbp2* | IDT | forward, 5'-GACCAGAGTG GGGTAGACGA-3', reverse, 5'-AAGGTTGGAAAG AAGCCCACA-3' | |
| Sequence-based reagent | qRT-PCR primers, mouse *Osal2* | IDT | forward, 5'-TCCTGACGAC CTCGTTTTGG-3', reverse, 5'-TCCTGACGAC CTCGTTTTGG-3' | |
| Sequence-based reagent | qRT-PCR primers, mouse *Ctss* | IDT | forward, 5'-ATCACTGCGGAAT TGCTAGTT-3', reverse, 5'-ACGACACACTT GGTTCCTCT-3' | |
| Sequence-based reagent | qRT-PCR primers, mouse *Laptm5* | IDT | forward, 5'-TCTCTGCCCCC TAAGACTCC-3', reverse, 5'-CCTGGTGGGG ATCACACTTC-3' | |
| Sequence-based reagent | qRT-PCR primers, mouse *Ctsc* | IDT | forward, 5'-CTGCTTTCCCT ACACAGCCA-3', reverse, 5'-ACGGAGGCAA TTCTCCCTTG-3' | |
| Sequence-based reagent | qRT-PCR primers, mouse *Slc11a1* | IDT | forward, 5'-CATCCAGCAA GCAAAGAGGC-3', reverse, 5'-TCCAGAAAGC CAGTAGGGGA-3' | |
| Sequence-based reagent | qRT-PCR primers, mouse *Ctsh* | IDT | forward, 5'-AGACCAAGGGA GGAACTGGT-3', reverse, 5'-GGTGGGCTTG TCGCTATTCA-3' | |
| Sequence-based reagent | qRT-PCR primers, mouse *Cd14* | IDT | forward, 5'-GAATTGGGCGAG AGAGGACT-3', reverse, 5'- TCCTGACGACCTCCGCTA AAACTTGGAGGGTCG-3' | |
| Sequence-based reagent | qRT-PCR primers, mouse *Dusp5* | IDT | forward, 5'-ACTTCAGACCAT CCCCAAGG-3', reverse, 5'-TGAGGTGCAA GGACTAGGTG-3' | |
| Sequence-based reagent | qRT-PCR primers, mouse *Flnc* | IDT | forward, 5'- AAAGAGCAAT GGAAGACGGC-3', reverse, 5'-CCACACATCA CATGCTGCTT-3' | |

*Continued on next page*

*Continued*

| Reagent type (species) or recourse | Designation | Source or reference | Identifier | Additional information |
|---|---|---|---|---|
| Sequence-based reagent | qRT-PCR primers, mouse *Myc* | IDT | forward, 5'-TCAGACACGG AGGAAAACGA-3', reverse, 5'-GTTCCTCCTC TGACGTTCCA-3' | |
| Sequence-based reagent | qRT-PCR primers, mouse *Rac2* | IDT | forward, 5'-CTTCCTGCCT GTTTTGGGTC-3', reverse, 5'-ACCTGAACTTG ACCTCGGAG-3' | |
| Commercial assay or kit | SuperScript First-Strand Synthesis System | Thermo Fisher Scientific | 1800051 | |
| Commercial assay or kit | iQSYBR Green Supermix | Bio-Rad | 1708880 | |
| Commercial assay or kit | Illumina TruSeq RNA Sample Prep Kit | Illumina | RS-122–2001 | |
| Chemical compound, drug | α-Bugarotoxin, Alexa Fluor 488 conjugate | Thermo Fisher Scientific | B13422 | |
| Chemical compound, drug | Fluoromyelin Red Fluorescent Myelin stain | Thermo Fisher Scientific | F34652 | |
| Software, algorithm | GraphPad Prism 7.0 | GraphPad Software | RRID:SCR_002798 | |
| Software, algorithm | ComBat | *Leek et al., 2012* | RRID:SCR_010974 | |
| Software, algorithm | Kallisto-Sleuth | Pachter lab, https://pachterlab.github.io/kallisto/ and https://pachterlab.github.io/sleuth/about | Kallisto: RRID:SCR_016582 sleuth: RRID:SCR_016883 | |
| Software, algorithm | Cytoscape | *Smoot et al., 2011* | RRID:SCR_003032 | |
| Software, algorithm | ClueGO | *Bindea et al., 2009* | RRID:SCR_005748 | |
| Software, algorithm | Bioquant Software | BIOQUANT Life Science | RRID:SCR_016423 | |

## Generation of transgenic mice expressing floxed wild type and ALS-linked mutations in FUS

cDNA encoding human FUS and its ALS-linked mutations (R514G and R521C) were amplified by PCR with a N-terminal hemagglutinin (HA) tag and cloned into Xho-I site of the MoPrp.Xho plasmid (ATCC#JHU-2). Open reading frames of the resulting plasmids were sequenced and confirmed. To generate 'floxed' transgene constructs, Not-I was used to liberate the transgene fragments and subcloned into vector containing two flanking loxP sites. 'Floxed' transgene constructs were excised with Cla-I and injected into the pronuclei of fertilized eggs to generate FUS transgenic mice (in C57Bl6/C3H hybrid background). Multiple founder lines with varying expression level of the transgenes were obtained. Lines of comparable wild type or mutant human FUS accumulation were selected for subsequent analysis. Mice were backcrossed to C57Bl6 for more than five generations

and used for analysis in this paper. Genotype primers used in this study are: 5'-GAG GAT TTC CCA GTG GAG GT-3' and 5'-CTC CAT CAA AGG GAC CTG AA-3'.

All studies were carried out under protocols approved by the Institutional Animal Care and Use Committee of the University of California, San Diego (UCSD) and the National University of Singapore (NUS), and were in compliance with Association for Assessment of Laboratory Animal Care guidelines for animal use. All studies were performed in such a manner as to minimize group size and animal suffering. The approved NUS protocol numbers are BR17-0928 and R16-0954.

## RT-QPCR

Total RNA from half a mouse cortex was isolated in Trizol (Invitrogen) and prepared for reverse transcription according to manufacturer's instructions. Real time quantitative PCR was performed on 40 ng of total cDNA using the iQSYBR Green supermix (Bio-Rad) with the iCycler iQ detection system according to manufacturer's instructions. Mus musculus ribosomal protein S9 (Rps9, NM_029767) and actin gamma subunit protein (Actg1, NM_009609) genes were also measured as endogenous references across all experimental conditions.

## Immunohistochemistry, motor neuron and axon quantification

Tissue preparation for immunohistochemistry was described previously (*Arnold et al., 2013*). In brief, anesthetized mice were transcardialy perfused with phosphate buffered saline (PBS), followed by 4% paraformaldehyde (PFA) in phosphate buffer for fixation. L5 roots of 3 ~ 5 animals per genotype and age point were collected and incubated in 2% osmium tetroxide in 0.05 M cacodylate buffer. The roots were subsequently washed, dehydrated and embedded in Epon (Electron Microscopy Sciences) for sectioning. 1 μm-thick cross sections were stained with 1% toluidine blue for 30 s. Both motor and sensory axons from L5 roots were quantified as described (*Arnold et al., 2013*).

Brains and spinal cords were post-fixed in 4% PFA for 2 hr, cryoprotected in 30% sucrose for over 24 hr and embedded in Tissue-Tek. For immunohistochemistry cryosections (30 μm) of fixed spinal cord and brain were rinsed in PBS, incubated in a blocking solution containing PBS, 0.5% Tween-20, 1.5% BSA for 1 hr at room temperature and transferred for an overnight incubation at room temperature in PBS, 0.3% Triton-X100 supplemented with the following primary antibodies: mouse anti-HA (Covance) at 1:5,000, anti-p62/SQSTMQ (Enzo) at 1:500, goat anti-ChAT (Chemicon) at 1:300, rabbit anti-GFAP (Dako) at 1:1000 and mouse anti-Iba1 at 1:1000. Primary antibodies were washed with PBS and then detected using donkey anti-rabbit Cy3, anti-mouse Cy3, anti-goat Cy3 (1:500) coupled secondary antibodies (Jackson ImmunoResearch). The secondary antibodies were washed with PBS and the spinal cord sections were either directly mounted or further incubated with a monoclonal antibody against neuronal nuclei marker, NeuN-Alexa488 (1:1,000, Chemicon) for 1.5 hr at room temperature. The sections were washed with PBS and mounted. Analysis was performed on a Nikon Eclipse laser scanning confocal microscope.

All lumbar spinal cord choline acetyl-transferase (ChAT) positive motor neurons were counted in the ventral horn of at least 25 sections per animal (in three mice per genotype). The total number of motor neurons counted was then divided by the number of sections.

Evaluation of muscle innervation at the neuromuscular junction (NMJ) was performed by immunohistochemistry on gastrocnemius. Floating sections (40 μm) were incubated in a blocking solution containing PBS, 0.5% Tween-20, 1.5% BSA for 4 hr at room temperature and then in PBS, 0.3% Triton-X100 overnight at room temperature with the polyclonal rabbit anti-synaptophysin antibody at 1:50 (Invitrogen). The sections were washed with PBS and then incubated first with donkey anti-rabbit Cy3 (Jackson ImmunoResearch) and α-Bungarotoxin-Alexa488 (Invitrogen) at 1:500 for 1 hr at room temperature and then with FluoroMyelin red (Invitrogen) at 1:300 for 30 min. The sections were further washed with PBS and mounted. Analysis was performed on a Nikon Eclipse laser scanning confocal microscope. A total of approximately 1000 neuromuscular junctions were counted from at least 10 sections of gastrocnemius. Individual NMJs were considered as innervated when colocalization between synaptophysin and α-Bungarotoxin staining was over 20%.

## RNA-seq library preparation and sequencing, and bioinformatics analysis

### Abundance quantification and differential expression calling

RNA quality was measured using the Agilent Bioanalyzer system. Samples with RIN (RNA integrity numbers) larger than 8.0 were used for RNA library preparation. Multiplex strand specific RNA-seq libraries were prepared from 30-day-old mouse spinal cord RNA and ASO-treated mouse spinal cord RNA using Illumina TruSeq RNA Sample Prep Kit and libraries were sequenced using Illumina HiSeq 4000 single-ended 50 bp sequencing. Read quantification was performed with Kallisto (0.44.0) (*Bray et al., 2016*) with parameters -b 50 –single -l 200 s 20 using ENSEMBL cDNA transcripts (release 91). Downstream differential gene expression calling was performed using Sleuth (0.28.1) (*Pimentel et al., 2017*). Quantified genes from each sample were annotated with a condition tag corresponding to the sample genotype. For each gene, Wald testing was performed on the condition parameter to obtain their respective FDR-corrected p-values. Significance for each gene was then established under a cutoff of FDR < 0.1.

### Diagnostic plot generation

Diagnostic plots (MA, Principal Components Analysis) were generated using the R statistical language's ggplot2 package. Expression data matrices were log-scaled before treatment with ComBat (*Leek et al., 2012*), to correct batch effect stemming from the different sequencing experiments. Principal Components analysis was then performed on the batch-corrected gene Transcripts Per Million (TPM) values generated from Kallisto-sleuth gene expression quantifications. The expression data matrices from the PCA where also used for hierarchical clustering and heatmap generation. From this, a subset corresponding to the top 2500 genes by ordered by PCA loading values across the first five principal components was derived. This sub-matrix was then Z-scaled and centered before biclustering and rendering with the superheat R library (*Barter and Yu, 2018*).

### Visualization of read coverage

Raw read data were mapped against the ENSEMBL mouse genome and associated annotations (release 91) using the STAR aligner package (2.4.2a) with parameters: sjdbOverhang = 49. Read coverage of the various datasets was calculated using bedtools genomecov (2.17.0, -bg -split) using the GViz genomic data visualization toolkit (*Harmston et al., 2015*) (1.20.0).

### Gene ontology analysis

Selected FDR ordered gene sets were isolated from the *Cnp*- heterozygous *FUS^WT^* vs homozygous *FUS^WT^* dataset. The genes were then subjected to gene ontology enrichment using the Cytoscape package ClueGO (*Bindea et al., 2009*) (2.3.0) with parameters ontology = Biological Process (GO_BiologicalProcess-GOA_23.02.2017_10h01), GO term fusion = True, GOLevels=(*Gao et al., 2017*; *Van Deerlin et al., 2008*). The resultant network was visualized in Cytoscape (*Smoot et al., 2011*) (3.5.0) under a Benjamini-Hochberg FDR corrected p<0.05 cutoff using the included organic layout. Clusters of interest for downstream analysis were then identified from the network manually.

### Cluster Gene Pathway Projection

Pathway projection of the GO-term cluster associated genes was performed as follows: For each selected cluster, we derived the set of all genes associated with all GO terms contained by the cluster. We then filtered this cluster gene set, removing all genes which were not differentially expressed in the non-transgenic vs heterozygous *FUS^WT^* analysis. The resultant gene list was then annotated with their respective foldchanges and submitted to the Reactome analyzer service for pathway enrichment. The resultant projection was then visualized using the Reactome pathway browser.

### Proportional expression analysis

OE and KD gene datasets were joined on their associated Ensembl gene ID. The resultant combined table was then subset into two subtables: The 'insignificant' subset was filtered for genes with FDR values above our specified cutoff (FDR > 0.01) in either OE or KD datasets. The other 'significant' table was created similarly but was instead constrained to genes with FDRs below the cutoff value in

both datasets. For each table, oppositely and similarly expressed genes were enumerated and the proportional distribution derived. A binomial test to determine a proportional shift was then performed, comparing the data against the null model (p(Opp)=0.5; p(Sim)=0.5).

RNA-seq data have been deposited in NCBI's Gene Expression Omnibus with the GEO series accession number GSE125125.

## Animal behavior and electrophysiology

### Resting electromyographic (EMG) recording

Animals were anesthetized with 2.5% isoflurane and the left hind limb shaved. To record EMG, two 30G platinum transcutaneous needle electrodes (Grass Technologies, An Astro-Med, Inc., West Warwick, Ri) were placed into the gastrocnemius muscle (distance between recording electrodes ~ 1 cm). Electrodes were connected to an active headstage (3110W Headstage, Warner Instruments LLS), recorded signal amplified using DP-311 differential amplifier (Warner Instruments LLS) and digitalized by the PowerLab 8/30 data acquisition system (AD Instruments, Inc., Colorado Springs, CO). Recorded signal was sampled at 20 kHz and stored in PC for analysis.

### Gait analysis

The footprint test was used to analyze the gait of the transgenic mice. The front and hind paws were coated with yellow and black non-toxic paints, respectively. The animals were then placed onto a 60 cm long, 10 cm wide runway with a fresh sheet of paper placed on the floor of the runway for each run. Each mouse had three runs and the runs were averaged. The center of each paw print was identified manually and the distances were measured as described.

## Autophagy assay

### Expression constructs

The constructs used in this study were PCR-cloned into the pEBFP2 expressing vector (addgene #14893), at the Nhe I and Age I sites using the following primers: 5' forward: ctaggctagcgccaccatggcctcaaacgattatac; 3' reverse: ctagaccggtggatacggcctctccctgcgat. The BFP was fused in frame to the C-terminus of the respective FUS wild type construct or its mutants (R514G and R521C). The pDest-mCherry-EGFP-hLC3b (Addgene# 22418) construct was supplied Dr. Paul Taylor, and the mCherry-GFP-LC3 vector was obtained from Dr. Terje Johansen.

### Cell culture studies

Neuro2a mouse neuroblastoma cell line was purchased from ATCC (ATCC CCL-131). Cells are fully tested for sterility from mycoplasma contamination prior to distribution by ATCC. Mycoplasma was tested quarterly and was negative. The species is also confirmed and authenticated by the Cytochrome oxidase 1 (CO1) barcode assay. Neuro2a cells were plated in a 24 well dish in DMEM media in the presence of 10% FBS, and transfected following the manufacturer's protocol. The media was changed to media without antibiotics, and 10 hr later the transfected cells were cultured in either complete media (CM) or nutrient limited media (NLM) for 8 hr. For autophagic induction studies, Neuro2A cells were fixed with 4% paraformaldehyde (PFA) for 21 min on ice, washed with PBS, and then mounted on Fluoromount G (Electron Microscopy Sciences). For p62 studies, transfected and treated cells were fixed with 4% PFA as described above, permeabilized with 0.20% Triton X-100 for 5 min on ice, and them immediately blocked with 5% BSA and goat serum for 1 hr. Cells were then incubated overnight in BFP (Abcam #32791, diluted to 1:400 in blocking serum) and p62 antibody (Enzo # BML-PW9860, diluted to 1:500 in blocking serum). The next day the primary antibody was removed and the cells were washed with PBS followed by 45 min incubation with goat anti-mouse Alexa 350 (Invitrogen, Cat#A21049, diluted to 1:100 in blocking serum) and goat anti-rabbit Alexa 594 (Invitrogen, A11037, diluted to 1:100 in blocking serum). Then the cells were stained with Syto13, washed in PBS, and mounted on Fluoromount G on glass slides.

### Autophagy assay and imaging

Cell counting was performed on a Nikon eclipse 80i microscope and $\geq$100 cells were counted for each experiment. Cells were scored as positive for autophagy induction when more than seven puncta were present. To determine the autophagic flux, Neuro2A cells co-transfected with mCherry-

GFP-LC3 and BFP or FUS-BFP and imaged using a Zeiss 780 confocal microscope, and $\geq$40 cells were imaged per experiment. Autophagosomes (yellow/yellow green puncta) and autolysosomes (red puncta) were counted for each cell, and the percentage of autophagosomes or autolysosomes calculated out of total puncta/cell. For p62 analysis, 10–30 images were obtained to visualize at least 60 transfected cells/experiment on a Ziess 780 confocal microscope. Cells with more than seven p62 puncta were scored as positive.

### Statistical analysis

We performed unpaired two-tailed t-test (non-parametric, Student's t test) or one-way ANOVA (using Bonferroni's multiple comparison test). All experiments were repeated at least three times.

### Supplementary information

Transgene insertion sites: nucleotide number 84,017,962 of chromosome 8 for line 101 of $FUS^{WT}$, nucleotide number 80,844,996 of chromosome 15 for line 136 of $FUS^{WT}$, nucleotide number 23,703,369 of chromosome 3 for line 135 of $FUS^{R521C}$, nucleotide number 40,067,721 of chromosome 17 for line 136 of $FUS^{R521C}$ (nucleotide number refers to NCBI137/mm9 mouse assembly in the UCSC genome browser, http://genome.ucsc.edu).

## Acknowledgements

We are grateful for the help of Janet Folmer (The John Hopkins University, Baltimore, MD), Ying Jones and Timothy Meerloo (University of California at San Diego, La Jolla, CA) for section and technical support for the electron microscope, Kristen Watanabe, Christie Duong, Erik Balinghasay, Kevin Clutario, Kevin Drenner, Sandra Lee, and Jihane Boubaker for technical assistance. This work was supported by grants to DWC from the NIH and Wellcome trust, to ARL from the NIH (R01 AG033082 and R01 NS041648), and to S-C L from the Swee Liew-Wadsworth Endowment fund, National University of Singapore (NUS), National Medical Research Council (NMRC/OFIRG/0001/2016 and NMRC/OFIRG/0042/2017) and Ministry of Education (MOE2016-T2-1-024), Singapore. S-C Ling dedicates this work to Sheue-Houy Tyan.

## Additional information

#### Competing interests

Don W Cleveland: Reviewing editor, *eLife*. The other authors declare that no competing interests exist.

#### Funding

| Funder | Grant reference number | Author |
| --- | --- | --- |
| National Medical Research Council | NMRC/OFIRG/0001/2016 | Shuo-Chien Ling |
| Ministry of Education - Singapore | MOE2016-T2-1-024 | Shuo-Chien Ling |
| National Institutes of Health | R01 AG033082 | Albert R La Spada |
| Wellcome Trust | | Christopher E Shaw |
| National Institutes of Health | R01 NS041648 | Albert R La Spada |

The funders had no role in study design, data collection and interpretation, or the decision to submit the work for publication.

#### Author contributions

Shuo-Chien Ling, Conceptualization, Resources, Data curation, Formal analysis, Supervision, Funding acquisition, Validation, Investigation, Visualization, Methodology, Writing—original draft, Project administration, Writing—review and editing; Somasish Ghosh Dastidar, Data curation, Funding

acquisition, Validation, Investigation, Visualization, Methodology, Writing—original draft, Writing—review and editing; Seiya Tokunaga, Wan Yun Ho, Data curation, Formal analysis, Validation, Investigation, Visualization, Methodology, Writing—review and editing; Kenneth Lim, Data curation, Software, Formal analysis, Validation, Investigation, Visualization, Methodology, Writing—original draft, Writing—review and editing; Hristelina Ilieva, Philippe A Parone, Sandrine Da Cruz, Conceptualization, Data curation, Formal analysis, Supervision, Validation, Investigation, Visualization, Methodology, Writing—original draft, Project administration, Writing—review and editing; Sheue-Houy Tyan, Formal analysis, Supervision, Investigation, Methodology, Writing—original draft, Writing—review and editing; Tsemay M Tse, Data curation, Validation, Investigation, Writing—review and editing; Jer-Cherng Chang, Anne Vetto, Shuying Sun, Data curation, Formal analysis, Validation, Investigation; Oleksandr Platoshyn, Data curation, Formal analysis, Investigation, Methodology, Writing—review and editing; Ngoc B Bui, Anh Bui, Data curation, Formal analysis, Validation, Investigation, Visualization; Melissa McAlonis-Downes, Joo Seok Han, Lino Tessarollo, Martin Marsala, Data curation, Formal analysis, Investigation; Debbie Swing, Investigation, Methodology; Katannya Kapeli, Gene W Yeo, Data curation, Formal analysis; Christopher E Shaw, Conceptualization, Funding acquisition, Investigation, Methodology, Project administration; Greg Tucker-Kellogg, Resources, Data curation, Software, Formal analysis, Supervision, Validation, Investigation, Visualization, Methodology, Writing—original draft, Writing—review and editing; Albert R La Spada, Clotilde Lagier-Tourenne, Data curation, Formal analysis, Supervision, Validation, Investigation, Visualization, Methodology, Writing—original draft; Don W Cleveland, Conceptualization, Data curation, Formal analysis, Supervision, Funding acquisition, Investigation, Methodology, Writing—original draft, Project administration, Writing—review and editing

### Author ORCIDs
Shuo-Chien Ling (ID) http://orcid.org/0000-0002-0300-8812
Lino Tessarollo (ID) http://orcid.org/0000-0001-6420-772X
Albert R La Spada (ID) https://orcid.org/0000-0001-6151-2964

### Ethics
Animal experimentation: All studies were carried out under protocols approved by the Institutional Animal Care and Use Committee of the University of California, San Diego (UCSD) and the National University of Singapore (NUS), and were in compliance with Association for Assessment of Laboratory Animal Care guidelines for animal use. All studies were performed in such a manner as to minimize group size and animal suffering. The approved NUS protocol numbers are BR17-0928 and R16-0954.

### Decision letter and Author response
Decision letter https://doi.org/10.7554/eLife.40811.038
Author response https://doi.org/10.7554/eLife.40811.039

## Additional files

### Supplementary files
• Supplementary file 1. GO and KEGG analysis of differentially expressed genes in the spinal cords of FUS-overexpression mice. Tab SF-1a: GO analysis: up-regulated differentially expressed genes (DEGs) in the spinal cords of FUS-overexpression (OE) mice Tab SF-1b: GO analysis: down-regulated differentially expressed genes (DEGs) in the spinal cords of FUS-overexpression (OE) mice Tab SF-1c: KEGG analysis: up-regulated differentially expressed genes (DEGs) in the spinal cords of FUS-overexpression (OE) mice Tab SF-1d: KEGG analysis: down-regulated differentially expressed genes (DEGs) in the spinal cords of FUS-overexpression (OE) mice
DOI: https://doi.org/10.7554/eLife.40811.032

• Supplementary file 2. GO analysis of differentially expressed genes in the spinal cords of FUS-overexpression and FUS-knockdown mice. Tab SF-2a: GO analysis: conversely regulated DEGs in the spinal cords of FUS-overexpression (OE) and FUS-knockdown (KD) mice (down-regulated in FUS-OE,

up-regulated in FUS-KD). Tab SF-2b: GO analysis: conversely regulated DEGs in the spinal cords of FUS-overexpression (OE) and FUS-knockdown (KD) mice (up-regulated in FUS-OE, down-regulated in FUS-KD) Tab SF-2c: GO analysis: common down-regulated DEGs in the spinal cords of FUS-over-expression (OE) and FUS-knockdown (KD) mice Tab SF-2d: GO analysis: common up-regulated DEGs in the spinal cords of FUS-overexpression (OE) and FUS-knockdown (KD) mice

DOI: https://doi.org/10.7554/eLife.40811.033

• Transparent reporting form
DOI: https://doi.org/10.7554/eLife.40811.034

## Data availability

RNA-seq data have been deposited in NCBI's Gene Expression Omnibus with the GEO series accession number GSE125125.

The following dataset was generated:

| Author(s) | Year | Dataset title | Dataset URL | Database and Identifier |
|---|---|---|---|---|
| Shuo-Chien Ling | 2019 | Overriding FUS autoregulation triggers gain-of-toxic dysfunctions in autophagy-lysosome axis and RNA metabolism | https://www.ncbi.nlm.nih.gov/geo/query/acc.cgi?acc=GSE125125 | NCBI Gene Expression Omnibu, GSE125125 |

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
