## [Decision Letter]

Thank you for submitting your article "Overriding FUS autoregulation triggers gain-of-toxic dysfunctions in RNA metabolism and autophagy-lysosome pathway" for consideration by *eLife*. Your article has been reviewed by three peer reviewers, and the evaluation has been overseen by a Reviewing Editor and James Manley as the Senior Editor. The following individuals involved in review of your submission have agreed to reveal their identity: Eric J Huang and Luc Dupuis.

The reviewers have discussed the reviews with one another and the Reviewing Editor has drafted this decision to help you prepare a revised submission.

Summary:

Mutations in FUS are causative of familial forms of ALS-FTD. The authors present a thorough examination of the consequences of expressing wild type and two mutant forms of human FUS in neurons, astrocytes and oligodendroglia of transgenic mice. The mice develop progressive motor deficits enhanced by disease mutation. The results show autoregulation of FUS expression (as has been observed previously) such that exogenous human FUS, either wild type or mutant, down-regulate endogenous mouse FUS. Consistent with these findings, heterozygous transgenic mice exhibit a very modest phenotype, whereas the homozygous or double heterozygous mice that express at supernormal levels develop more aggressive disease. Interestingly, CNS-directed expression of wild type human FUS can rescue the perinatal lethality in Fus KO mice (at least up to 40 days after birth) providing the first evidence that human FUS can functionally replace mouse Fus and showing that CNS expression of Fus is essential. Transcriptomic analyses reveal differentially expressed genes implicated in the lysosomal pathway, MAPK pathway and neuronal functions, suggesting a dose-dependent effect of exogenous FUS. The transcriptomic alterations driven by Fus depletion or FUS over-expression are different, and evidence is presented that degeneration is primarily driven by a gain-of-function. Finally, the authors provide in vitro evidence that overexpression of human FUS (wild type and mutant alike) can disrupt autophagy.

Essential revisions:

• Transcriptomic analyses were conducted at P30, an age by which these mice exhibit advanced neurodegeneration and mortality. This raises a concern that the changes observed are dominated by secondary changes that may be only distantly and indirectly related to the primary insult. The authors should perform further analysis at early time points, which would ideally involve RNAseq. If RNAseq at early time points is not feasible, representative genes identified at late time points (e.g. "long" genes) should be validated individually by RT-PCR.

• The authors indicate that they have performed gene ontology (GO) analyses (subsection “Altered RNA processing functions with elevated FUS expression”, first paragraph) and identified several GO categories significantly altered in homozygous transgenic mice. It is unclear whether the GO categories indicated were highlighted in a biased or unbiased way. An unbiased presentation of GO categories should be provided, for example, with a graph indicating the top 10 GO categories for up- or down-regulated genes, and the specific ID of the genes in each category. Similar GO analyses should be performed for genes in Figure 7F.

• It is unclear the extent to which altered transcripts represent direct targets of FUS (as will be indicated by existing CLIP data). This analysis, including any bias in directionality, may be informative about direct vs. indirect effects of altered FUS levels.

• The mice develop not only a progressive motor phenotype but also significant sensory deficits. E.g. the R521G mice show significant sensory loss, without significant changes in motor neuron loss, which differs from human, FUS-related ALS. Is this due to unnatural (i.e. Prnp-driven) expression pattern? This discrepancy should be highlighted and discussed.

• The approach to estimating upper motor neuron counts is not appropriate. Retrograde labeling is the method of choice to quantify corticospinal neurons in mice. Moreover, it is likely that NeuN/SMI32 positive neurons in the frontal cortex are poorly representative of corticospinal neurons. The reviewers recommend using retrograde labeling or, since this claim is not essential to the manuscript, simply removing this data.

[Editors' note: further revisions were requested prior to acceptance, as described below.]

Thank you for resubmitting your work entitled "Overriding FUS autoregulation in mice triggers gain-of-toxic dysfunctions in RNA metabolism and autophagy-lysosome axis" for further consideration at *eLife*. Your revised article has been favorably evaluated by James Manley (Senior Editor), a Reviewing Editor, and two reviewers.

The manuscript has been improved but there is consensus that some remaining issues must be addressed before acceptance. A summary of the two concerns follows:

1) Interpretation of RNA data: A comparison of transcription changes at P14 and P30 suggests that the massive transcription changes at late stage are dominated by the effects of neurodegeneration. The authors should rephrase the results by underlining that FUS over-expression does not induce widespread transcription changes (P14 data), but does lead to toxicity, and at late stages this involves alterations in RNA metabolism and autophagy. It is of great interest that these pathways are changed, but there is no evidence these are primary changes and therefore this should not be asserted on the basis of these data.

2) Rescue of FUS KO: The data support the assertion that "Human FUS expression can rescue mouse FUS KO", but the authors should not make the extra claim that this is due to CNS expression, as they do not show this. Nevertheless, this finding is still of interest, especially considering that recent data from the Talbot group has shown that this is not the case for TDP-43.

---

## [Author Response]

Essential revisions:• Transcriptomic analyses were conducted at P30, an age by which these mice exhibit advanced neurodegeneration and mortality. This raises a concern that the changes observed are dominated by secondary changes that may be only distantly and indirectly related to the primary insult. The authors should perform further analysis at early time points, which would ideally involve RNAseq. If RNAseq at early time points is not feasible, representative genes identified at late time points (e.g. "long" genes) should be validated individually by RT-PCR.

We thank the editor and reviewers for the helpful comment and suggestions. We have performed the qRT-PCR analysis on P14 mice, where mice display no apparent phenotype (Figure 3E). At P14, the majority of deregulated mRNAs of FUS-over-expressing mice at P30 showed normal expression levels as those of control mice. The new data are included in the new Figure 6—figure supplement 3.

Although the data suggest a correlation between deregulated RNA metabolism with disease phenotype, we agree with the editor and reviewers that ideally, additional RNA-seq experiments should be performed at different time points, such as P16, P18, etc., to track disease progression. However, as the editor and reviewers pointed out, this is not feasible for the time given for the revision. Instead, we focused on the analyses outlined below. The new analyses provide new insight on the potential disease mechanisms caused by FUS dysfunctions.

• The authors indicate that they have performed gene ontology (GO) analyses (subsection “Altered RNA processing functions with elevated FUS expression”, first paragraph) and identified several GO categories significantly altered in homozygous transgenic mice. It is unclear whether the GO categories indicated were highlighted in a biased or unbiased way. An unbiased presentation of GO categories should be provided, for example, with a graph indicating the top 10 GO categories for up- or down-regulated genes, and the specific ID of the genes in each category. Similar GO analyses should be performed for genes in Figure 7F.

We thank the editor and reviewers for the suggestion and apologize for the oversight and lack of clarity on this part of analysis. Differentially expressed genes (DEGs) were categorized with unbiased gene ontology (GO) biological process analyses and mapped onto KEGG (Kyoto Encyclopedia of Genes and Genomes) pathways. The top categories of unbiased GO analyses and KEGG pathway analyses were included in new Figure 6—figure supplement 1 and Supplementary file 1A and 1B (for GO analyses) and Supplementary file 1C and 1D(for KEGG analyses). As evident from those new supplementary materials, GO analysis showed distinct themes of up and down-regulated genes which we highlight in the text: top up-regulated GO categories are involved in defense response, innate immune response and regulation of immune system process, whereas top down-regulated GO categories are primarily metabolic, including steroid biosynthesis, cellular nitrogen compound metabolism process and cholesterol biosynthesis. KEGG analysis showed that the up-regulated genes are enriched in the KEGG pathways involved in lysosomes, antigen processing and presentation, cytokine-cytokine receptor interactions, MAPK or p53 signaling pathways, and ECM-receptor interactions, whereas KEGG pathways of the down-regulated genes were involved in steroid biosynthesis and terpenoid backbone biosynthesis.

In addition, we also performed the GO analyses on the genes that are either co-regulated (similarly perturbed) or oppositely-regulated (conversely perturbed) by FUS overexpression (OE) and knockdown (KD). The analyses revealed that distinct sets of genes were affected when FUS-OE and FUS-KD conditions were compared. Specifically, (i) in the OE-down/KD-up group, mitochondrial perspiratory chain and ribose phosphate metabolic process are affected; (ii) in the OE-down/KD-up group, stem cell population maintenance and cytokine-mediated signaling pathway are affected; (iii) in the OE down/KD down group, central nervous system myelination, regulation of synaptic plasticity affected; and (iv) in the OE up/KD up, innate immune response and inflammatory response are affected. The findings suggest that (1) similar biological process, such as synaptic functions and immune response, can be affected in the similar fashion by FUS-OE and FUS-KD, and conversely, (2) FUS-OE and FUS-KD can have opposite effects on other biological pathways, such as mitochondrial functions. These alterations of biological processes caused by varying FUS levels would aid to clarify the potential FUS-mediated pathogenic mechanisms. The new data is included in the revised Figure 7 and the GO analyses for each group are summarized in Supplementary file 2(with 4 additional tables).

• It is unclear the extent to which altered transcripts represent direct targets of FUS (as will be indicated by existing CLIP data). This analysis, including any bias in directionality, may be informative about direct vs. indirect effects of altered FUS levels.

We thank the editor and reviewers for the insightful comment and suggestions. It should be noted that mRNA binding sites of FUS are noisier when compared with TDP-43, another ALS-FTD linked RNA-binding proteins. This may in part be due to the known co-transcriptionally deposition of FUS on newly synthesized pre-mRNAs. Nevertheless, based on the known FUS-binding sites (Lagier-Tourenne et al., 2012), 15.1% of up-regulated genes (337 out of 2228 genes) and 26.1% of down-regulated genes (483 out of 1853 genes) are bound by FUS. On the other hand, 23% of oppositely-regulated mRNAs (conversely perturbed category: expression changes are in opposing directions in FUS-OE and knockdown (KD) conditions, 199 out of 862 genes) and 31% of co-regulated mRNAs (similarly perturbed category: the directions of expression change are the same in both FUS-OE and FUS-KD conditions, 158 out of 507 genes) are bound by FUS. In the conversely perturbed category, FUS binds to 27% and 17% of mRNAs that are either down- or up-regulated in FUS-OE conditions, whereas FUS binds to 31% of the down-regulated mRNAs (69 out of 222) and 31% up-regulated mRNAs (89 out of 285) in the similarly perturbed category. We have added the information about FUS-mRNA binding sites in the text and the figures (Figure 7D and Figure 7—figure supplement 1B).

• The mice develop not only a progressive motor phenotype but also significant sensory deficits. E.g. the R521G mice show significant sensory loss, without significant changes in motor neuron loss, which differs from human, FUS-related ALS. Is this due to unnatural (i.e. Prnp-driven) expression pattern? This discrepancy should be highlighted and discussed.

We thank the editor and reviewers for the insightful comment. Although R521C mutation is one of the most frequent FUS mutations in ALS, expression of R521C in mice did not produce motor axon loss, but rather sensory axon loss, which differs from the FUS-related ALS in human. By contrast, wild-type and R514G transgene produce both the motor and sensory axon degeneration. Thus, it is formally possible that the observed sensory deficits may be caused by transgene expression driven by the murine prion promoter. However, we have previously showed that GEM bodies in the spinal cord motor neurons of FUS^R521C^ mice were significantly reduced (Sun et al., 2015), an indication of reduced SMN (survival motor neuron) function. This is reminiscent of the notion that altered circuit function could underlie the neurodegeneration as in the case of spinal muscular atrophy (SMA) (Mentis et al., 2011), in which the dysfunction in cholinergic sensory neurons and interneurons could lead to motoneuron degeneration (Imlach et al., 2012, Lotti et al., 2012). Further studies are required to investigate the possibilities of altered motor-sensory circuit in FUS pathogenesis. This part of text was added to the Discussion section.

• The approach to estimating upper motor neuron counts is not appropriate. Retrograde labeling is the method of choice to quantify corticospinal neurons in mice. Moreover, it is likely that NeuN/SMI32 positive neurons in the frontal cortex are poorly representative of corticospinal neurons. The reviewers recommend using retrograde labeling or, since this claim is not essential to the manuscript, simply removing this data.

We agree with the helpful comment and suggestion and have removed this part of data and claim.

[Editors' note: further revisions were requested prior to acceptance, as described below.]

The manuscript has been improved but there is consensus that some remaining issues must be addressed before acceptance. A summary of the two concerns follows:1) Interpretation of RNA data: A comparison of transcription changes at P14 and P30 suggests that the massive transcription changes at late stage are dominated by the effects of neurodegeneration. The authors should rephrase the results by underlining that FUS over-expression does not induce widespread transcription changes (P14 data), but does lead to toxicity, and at late stages this involves alterations in RNA metabolism and autophagy. It is of great interest that these pathways are changed, but there is no evidence these are primary changes and therefore this should not be asserted on the basis of these data.

We thank the editor and reviewers for the helpful comment and suggestions. We have edited the paragraph to include these changes, specifically in the subsection “Altered RNA processing functions with elevated FUS expression”.

2) Rescue of FUS KO: The data support the assertion that "Human FUS expression can rescue mouse FUS KO", but the authors should not make the extra claim that this is due to CNS expression, as they do not show this. Nevertheless, this finding is still of interest, especially considering that recent data from the Talbot group has shown that this is not the case for TDP-43.

We thank the editor and reviewers for the helpful comment and suggestions. We have removed the claim.